# Online Corrupted User Detection and Regret Minimization

**Zhiyong Wang**
The Chinese University of Hong Kong
zywang21@cse.cuhk.edu.hk

**Jize Xie**
Shanghai Jiao Tong University
xjzzjl@sjtu.edu.cn

**Tong Yu**
Adobe Research
worktongyu@gmail.com

**Shuai Li**[*]
Shanghai Jiao Tong University
shuaili8@sjtu.edu.cn

**John C.S. Lui**
The Chinese University of Hong Kong
cslui@cse.cuhk.edu.hk

## Abstract

In real-world online web systems, multiple users usually arrive sequentially into the system. For applications like click fraud and fake reviews, some users can maliciously perform corrupted (disrupted) behaviors to trick the system. Therefore, it is crucial to design efficient online learning algorithms to robustly learn from potentially corrupted user behaviors and accurately identify the corrupted users in an online manner. Existing works propose bandit algorithms robust to adversarial corruption. However, these algorithms are designed for a single user, and cannot leverage the implicit social relations among multiple users for more efficient learning. Moreover, none of them consider how to detect corrupted users online in the multiple-user scenario. In this paper, we present an important online learning problem named LOCUD to learn and utilize unknown user relations from disrupted behaviors to speed up learning, and identify the corrupted users in an online setting. To robustly learn and utilize the unknown relations among potentially corrupted users, we propose a novel bandit algorithm RCLUB-WCU. To detect the corrupted users, we devise a novel online detection algorithm OCCUD based on RCLUB-WCU's inferred user relations. We prove a regret upper bound for RCLUB-WCU, which asymptotically matches the lower bound with respect to $T$ up to logarithmic factors, and matches the state-of-the-art results in degenerate cases. We also give a theoretical guarantee for the detection accuracy of OCCUD. With extensive experiments, our methods achieve superior performance over previous bandit algorithms and high corrupted user detection accuracy.

## 1 Introduction

In real-world online recommender systems, data from many users arrive in a streaming fashion [4, 15, 2, 7, 35, 27, 26]. There may exist some corrupted (malicious) users, whose behaviors (*e.g.*, click, rating) can be adversarially corrupted (disrupted) over time to fool the system [29, 30, 12, 10, 9]. These corrupted behaviors could disrupt the user preference estimations of the algorithm. As a result, the system would easily be misled and make sub-optimal recommendations [14, 23, 7, 41],

---

[*]Corresponding author.

37th Conference on Neural Information Processing Systems (NeurIPS 2023).

which would hurt the user experience. Therefore, it is essential to design efficient online learning algorithms to robustly learn from potentially disrupted behaviors and detect corrupted users in an online manner.

There exist some works on bandits with adversarial corruption [29, 9, 22, 5, 12, 16]. However, they have the following limitations. First, existing algorithms are initially designed for robust online preference learning of a single user. In real-world scenarios with multiple users, they cannot robustly infer and utilize the implicit user relations for more efficient learning. Second, none of them consider how to identify corrupted users online in the multiple-user scenario. Though there also exist some works on corrupted user detection [34, 6, 39, 28, 13], they all focus on detection with *known* user information in an offline setting, thus can not be applied to do online detection from bandit feedback.

To address these limitations, we propose a novel bandit problem "*Learning and Online Corrupted Users Detection from bandit feedback*" (LOCUD). To model and utilize the relations among users, we assume there is an *unknown* clustering structure over users, where users with similar preferences lie in the same cluster [8, 19, 21]. The agent can infer the clustering structure to leverage the information of similar users for better recommendations. Among these users, there exists a small fraction of corrupted users. They can occasionally perform corrupted behaviors to fool the agent [12, 29, 30, 9] while mimicking the behaviors of normal users most of the time to make themselves hard to discover. The agent not only needs to learn the *unknown* user preferences and relations robustly from potentially disrupted feedback, balance the exploration-exploitation trade-off to maximize the cumulative reward, but also needs to detect the corrupted users online from bandit feedback.

The LOCUD problem is very challenging. First, the corrupted behaviors would cause inaccurate user preference estimations, which could lead to erroneous user relation inference and sub-optimal recommendations. Second, it is nontrivial to detect corrupted users online since their behaviors are dynamic over time (sometimes regular while sometimes corrupted), whereas, in the offline setting, corrupted users' information can be fully represented by static embeddings and the existing approaches [18, 32] can typically do binary classifications offline, which are not adaptive over time.

We propose a novel learning framework composed of two algorithms to address these challenges.

**RCLUB-WCU.** To robustly estimate user preferences, learn the unknown relations from potentially corrupted behaviors, and perform high-quality recommendations, we propose a novel bandit algorithm "*Robust CLUstering of Bandits With Corrupted Users*" (RCLUB-WCU), which maintains a dynamic graph over users to represent the learned clustering structure, where users linked by edges are inferred to be in the same cluster. RCLUB-WCU adaptively deletes edges and recommends arms based on aggregated interactive information in clusters. We do the following to ensure robust clustering structure learning. (i) To relieve the estimation inaccuracy caused by disrupted behaviors, we use weighted ridge regressions for robust user preference estimations. Specifically, we use the inverse of the confidence radius to weigh each sample. If the confidence radius associated with user $i_t$ and arm $a_t$ is large at $t$, the learner is quite uncertain about the estimation of $i_t$'s preference on $a_t$, indicating the sample at $t$ is likely to be corrupted. Therefore, we use the inverse of the confidence radius to assign minor importance to the possibly disrupted samples when doing estimations. (ii) We design a robust edge deletion rule to divide the clusters by considering the potential effect of corruptions, which, together with (i), can ensure that after some interactions, users in the same connected component of the graph are in the same underlying cluster with high probability.

**OCCUD.** To detect corrupted users online, based on the learned clustering structure of RCLUB-WCU, we devise a novel algorithm named "*Online Cluster-based Corrupted User Detection*" (OCCUD). At each round, we compare each user's non-robustly estimated preference vector (by ridge regression) and the robust estimation (by weighted regression) of the user's inferred cluster. If the gap exceeds a carefully-designed threshold, we detect this user as corrupted. The intuitions are as follows. With misleading behaviors, the non-robust preference estimations of corrupted users would be far from ground truths. On the other hand, with the accurate clustering of RCLUB-WCU, the robust estimations of users' inferred clusters should be close to ground truths. Therefore, for corrupted users, their non-robust estimates should be far from the robust estimates of their inferred clusters.

We summarize our contributions as follows.
• We present a novel online learning problem LOCUD, where the agent needs to (i) robustly learn and leverage the unknown user relations to improve online recommendation qualities under the disruption of corrupted user behaviors; (ii) detect the corrupted users online from bandit feedback.

• We propose a novel online learning framework composed of two algorithms, RCLUB-WCU and OCCUD, to tackle the challenging LOCUD problem. RCLUB-WCU robustly learns and utilizes the unknown social relations among potentially corrupted users to efficiently minimize regret. Based on RCLUB-WCU's inferred user relations, OCCUD accurately detects corrupted users online.
• We prove a regret upper bound for RCLUB-WCU, which matches the lower bound asymptotically in $T$ up to logarithmic factors and matches the state-of-the-art results in several degenerate cases. We also give a theoretical performance guarantee for the online detection algorithm OCCUD.
• Experiments on both synthetic and real-world data clearly show the advantages of our methods.

## 2 Related Work

Our work is related to bandits with adversarial corruption and bandits leveraging user relations.

The work [29] first studies stochastic bandits with adversarial corruption, where the rewards are corrupted with the sum of corruption magnitudes in all rounds constrained by the *corruption level $C$*. They propose a robust elimination-based algorithm. The paper [9] proposes an improved algorithm with a tighter regret bound. The paper [22] first studies stochastic linear bandits with adversarial corruptions. To tackle the contextual linear bandit setting where the arm set changes over time, the work [5] proposes a variant of the OFUL [1] that achieves a sub-linear regret. A recent work [12] proposes the CW-OFUL algorithm that achieves a nearly optimal regret bound. All these works focus on designing robust bandit algorithms for a single user; none consider how to robustly learn and leverage the implicit relations among potentially corrupted users for more efficient learning. Moreover, none of them consider how to online detect corrupted users in the multiple-user case.

Some works study how to leverage user relations to accelerate the bandit learning process in the multiple-user case. The work [38] utilizes a *known* user adjacency graph to share context and payoffs among neighbors. To adaptively learn and utilize *unknown* user relations, the paper [8] proposes the clustering of bandits (CB) problem where there is an *unknown* user clustering structure to be learned by the agent. The work [20] uses collaborative effects on items to guide the clustering of users. The paper [19] studies the CB problem in the cascading bandit setting. The work [21] considers the setting where users in the same cluster share both the same preference and the same arrival rate. The paper [25] studies the federated CB problem, considering privacy and communication issues. All these works only consider utilizing the relations among normal users; none of them consider how to robustly learn the user relations from potentially disrupted behaviors, thus would easily be misled by corrupted users. Also, none of them consider how to detect corrupted users from bandit feedback.

To the best of our knowledge, this is the first work to study the problem to (i) learn the unknown user relations and preferences from potentially corrupted feedback, and leverage the learned relations to speed up learning; (ii) adaptively detect the corrupted users online from bandit feedback.

## 3 Problem Setup

This section formulates the problem of "*Learning and Online Corrupted Users Detection from bandit feedback*" (LOCUD) (illustrated in Fig.1). We denote $\|\boldsymbol{x}\|_{\boldsymbol{M}} = \sqrt{\boldsymbol{x}^\top \boldsymbol{M} \boldsymbol{x}}$, $[m] = \{1, \ldots, m\}$, number of elements in set $\mathcal{A}$ as $|\mathcal{A}|$.

In LOCUD, there are $u$ users, which we denote by set $\mathcal{U} = \{1, 2, \ldots, u\}$. Some of them are corrupted users, denoted by set $\tilde{\mathcal{U}} \subseteq \mathcal{U}$. These corrupted users, on the one hand, try to mimic normal users to make themselves hard to detect; on the other hand, they can occasionally perform corrupted behaviors to fool the agent into making sub-optimal decisions. Each user $i \in \mathcal{U}$, no matter a normal one or corrupted one, is associated with a (possibly mimicked for corrupted users) preference feature vector $\boldsymbol{\theta}_i \in \mathbb{R}^d$ that is *unknown* and $\|\boldsymbol{\theta}_i\|_2 \leq 1$. There is an underlying clustering structure among all the users representing the similarity of their preferences, but it is *unknown* to the agent and needs to be learned via interactions. Specifically, the set of users $\mathcal{U}$ can be partitioned into $m$ ($m \ll u$) clusters, $V_1, V_2, \ldots V_m$, where $\cup_{j \in [m]} V_j = \mathcal{U}$, and $V_j \cap V_{j'} = \emptyset$, for $j \neq j'$. Users in the same cluster have the same preference feature vector, while users in different clusters have different preference vectors. We use $\boldsymbol{\theta}^j$ to denote the common preference vector shared by users in the $j$-th cluster $V_j$, and use $j(i)$ to denote the index of cluster user $i$ belongs to (*i.e.*, $i \in V_{j(i)}$). For

any two users $k, i \in \mathcal{U}$, if $k \in V_{j(i)}$, then $\boldsymbol{\theta}_k = \boldsymbol{\theta}^{j(i)} = \boldsymbol{\theta}_i$; otherwise $\boldsymbol{\theta}_k \neq \boldsymbol{\theta}_i$. We assume the arm set $\mathcal{A} \subseteq \mathbb{R}^d$ is finite. Each arm $a \in \mathcal{A}$ is associated with a feature vector $\boldsymbol{x}_a \in \mathbb{R}^d$ with $\|\boldsymbol{x}_a\|_2 \leq 1$.

The learning process of the agent is as follows. At each round $t \in [T]$, a user $i_t \in \mathcal{U}$ comes to be served, and the learning agent receives a set of arms $\mathcal{A}_t \subseteq \mathcal{A}$ to choose from. The agent infers the cluster $V_t$ that user $i_t$ belongs to based on the interaction history, and recommends an arm $a_t \in \mathcal{A}_t$ according to the aggregated information gathered in the cluster $V_t$. After receiving the recommended arm $a_t$, a normal user $i_t$ will give a random reward with expectation $\boldsymbol{x}_{a_t}^\top \boldsymbol{\theta}_{i_t}$ to the agent.

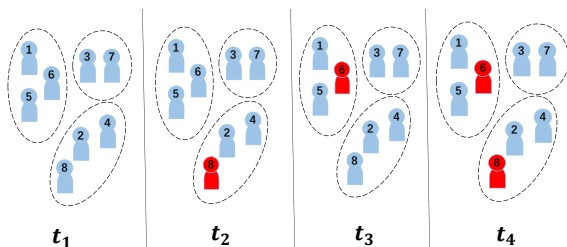

Figure 1: Illustration of LOCUD. The *unknown* user relations are represented by dotted circles, *e.g.*, user 3, 7 have similar preferences and thus can be in the same user segment (*i.e.*, cluster). Users 6 and 8 are corrupted users with dynamic behaviors over time (*e.g.*, for user 8, the behaviors are normal at $t_1$ and $t_3$ (blue), but are adversarially corrupted at $t_2$ and $t_4$ (red)[29, 12]), making them hard to be detected online. The agent needs to learn user relations to utilize information among similar users to speed up learning, and detect corrupted users 6, 8 online from bandit feedback.

To model the behaviors of corrupted users, following [29, 9, 5, 12], we assume that they can occasionally corrupt the rewards to mislead the agent into recommending sub-optimal arms. Specifically, at each round $t$, if the current served user is a corrupted user (i.e., $i_t \in \tilde{\mathcal{U}}$), the user can corrupt the reward by $c_t$. In summary, we model the reward received by the agent at round $t$ as

$$r_t = \boldsymbol{x}_{a_t}^\top \boldsymbol{\theta}_{i_t} + \eta_t + c_t \,,$$

where $c_t = 0$ if $i_t$ is a normal user, (*i.e.*, $i_t \notin \tilde{\mathcal{U}}$), and $\eta_t$ is 1-sub-Gaussian random noise.

As the number of corrupted users is usually small, and they only corrupt the rewards occasionally with small magnitudes to make themselves hard to detect, we assume the sum of corruption magnitudes in all rounds is upper bounded by the *corruption level* $C$, *i.e.*, $\sum_{t=1}^T |c_t| \leq C$ [29, 9, 5, 12].

We assume the clusters, users, and items satisfy the following assumptions. Note that all these assumptions basically follow the settings from classical works on clustering of bandits [8, 19, 25, 36].

**Assumption 1** (Gap between different clusters)**.** *The gap between any two preference vectors for different clusters is at least an unknown positive constant $\gamma$*

$$\left\|\boldsymbol{\theta}^j - \boldsymbol{\theta}^{j'}\right\|_2 \geq \gamma > 0 \,, \forall j, j' \in [m] \,, j \neq j' \,.$$

**Assumption 2** (Uniform arrival of users)**.** *At each round $t$, a user $i_t$ comes uniformly at random from $\mathcal{U}$ with probability $1/u$, independent of the past rounds.*

**Assumption 3** (Item regularity)**.** *At each round $t$, the feature vector $\boldsymbol{x}_a$ of each arm $a \in \mathcal{A}_t$ is drawn independently from a fixed unknown distribution $\rho$ over $\{\boldsymbol{x} \in \mathbb{R}^d : \|\boldsymbol{x}\|_2 \leq 1\}$, where $\mathbb{E}_{\boldsymbol{x} \sim \rho}[\boldsymbol{x}\boldsymbol{x}^\top]$'s minimal eigenvalue $\lambda_x > 0$. At $\forall t$, for any fixed unit vector $\boldsymbol{z} \in \mathbb{R}^d$, $(\boldsymbol{\theta}^\top \boldsymbol{z})^2$ has sub-Gaussian tail with variance no greater than $\sigma^2$.*

Let $a_t^* \in \arg\max_{a \in \mathcal{A}_t} \boldsymbol{x}_a^\top \boldsymbol{\theta}_{i_t}$ denote an optimal arm with the highest expected reward at round $t$. One objective of the learning agent is to minimize the expected cumulative regret

$$R(T) = \mathbb{E}\left[\sum_{t=1}^T (\boldsymbol{x}_{a_t^*}^\top \boldsymbol{\theta}_{i_t} - \boldsymbol{x}_{a_t}^\top \boldsymbol{\theta}_{i_t})\right] \,. \tag{1}$$

Another objective is to detect corrupted users online accurately. Specifically, at round $t$, the agent will give a set of users $\tilde{\mathcal{U}}_t$ as the detected corrupted users, and we want $\tilde{\mathcal{U}}_t$ to be as close to the ground-truth set of corrupted users $\tilde{\mathcal{U}}$ as possible.

## 4   Algorithms

This section introduces our algorithms RCLUB-WCU (Algo.1) and OCCUD (Algo.2). RCLUB-WCU robustly learns the unknown user clustering structure and preferences from corrupted feed-

back, and leverages the cluster-based information to accelerate learning. Based on the clustering structure learned by RCLUB-WCU, OCCUD can accurately detect corrupted users online.

---

**Algorithm 1** RCLUB-WCU

---

1: **Input:** Regularization parameter $\lambda$, confidence radius parameter $\beta$, threshold parameter $\alpha$, edge deletion parameter $\alpha_1$, $f(T) = \sqrt{(1 + \ln(1 + T))/(1 + T)}$.
2: **Initialization:** $\boldsymbol{M}_{i,0} = \boldsymbol{0}_{d \times d}, \boldsymbol{b}_{i,0} = \boldsymbol{0}_{d \times 1}, \tilde{\boldsymbol{M}}_{i,0} = \boldsymbol{0}_{d \times d}, \tilde{\boldsymbol{b}}_{i,0} = \boldsymbol{0}_{d \times 1}, T_{i,0} = 0 , \forall i \in \mathcal{U};$
   A complete graph $G_0 = (\mathcal{U}, E_0)$ over $\mathcal{U}$.
3: **for all** $t = 1, 2, \ldots, T$ **do**
4:    Receive the index of the current served user $i_t \in \mathcal{U}$, get the feasible arm set at this round $\mathcal{A}_t$.
5:    Determine the connected components $V_t$ in the current maintained graph $G_{t-1} = (\mathcal{U}, E_{t-1})$ such that $i_t \in V_t$.
6:    Calculate the robustly estimated statistics for the cluster $V_t$:
      $\boldsymbol{M}_{V_t,t-1} = \lambda \boldsymbol{I} + \sum_{i \in V_t} \boldsymbol{M}_{i,t-1} , \boldsymbol{b}_{V_t,t-1} = \sum_{i \in V_t} \boldsymbol{b}_{i,t-1} , \hat{\boldsymbol{\theta}}_{V_t,t-1} = \boldsymbol{M}_{V_t,t-1}^{-1} \boldsymbol{b}_{V_t,t-1} ;$
7:    Select an arm $a_t$ with largest UCB index in Eq.(3) and receive the corresponding reward $r_t$;
8:    Update the statistics for robust estimation of user $i_t$:
      $\boldsymbol{M}_{i_t,t} = \boldsymbol{M}_{i_t,t-1} + w_{i_t,t-1} \boldsymbol{x}_{a_t} \boldsymbol{x}_{a_t}^\top , \boldsymbol{b}_{i_t,t} = \boldsymbol{b}_{i_t,t-1} + w_{i_t,t-1} r_t \boldsymbol{x}_{a_t} , T_{i_t,t} = T_{i_t,t-1} + 1 ,$
      $\boldsymbol{M}'_{i_t,t} = \lambda \boldsymbol{I} + \boldsymbol{M}_{i_t,t}, \hat{\boldsymbol{\theta}}_{i_t,t} = \boldsymbol{M}'^{-1}_{i_t,t} \boldsymbol{b}_{i_t,t}, w_{i_t,t} = \min\{1, \alpha/\|\boldsymbol{x}_{a_t}\|_{\boldsymbol{M}'^{-1}_{i_t,t}}\} ;$
9:    Keep robust estimation statistics of other users unchanged:
      $\boldsymbol{M}_{\ell,t} = \boldsymbol{M}_{\ell,t-1}, \boldsymbol{b}_{\ell,t} = \boldsymbol{b}_{\ell,t-1}, T_{\ell,t} = T_{\ell,t-1} , \hat{\boldsymbol{\theta}}_{\ell,t} = \hat{\boldsymbol{\theta}}_{\ell,t-1},$ for all $\ell \in \mathcal{U}, \ell \neq i_t;$
10:   Delete the edge $(i_t, \ell) \in E_{t-1}$, if

$$\left\| \hat{\boldsymbol{\theta}}_{i_t,t} - \hat{\boldsymbol{\theta}}_{\ell,t} \right\|_2 \geq \alpha_1 \big( f(T_{i_t,t}) + f(T_{\ell,t}) + \alpha C \big) ,$$

   and get an updated graph $G_t = (\mathcal{U}, E_t);$
11:   Use the OCCUD Algorithm (Algo.2) to detect the corrupted users.
12: **end for**

---

## 4.1  RCLUB-WCU

The corrupted behaviors may cause inaccurate preference estimations, leading to erroneous relation inference and sub-optimal decisions. In this case, how to learn and utilize unknown user relations to accelerate learning becomes non-trivial. Motivated by this, we design RCLUB-WCU as follows.

**Assign the inferred cluster $V_t$ for user $i_t$.** RCLUB-WCU maintains a dynamic undirected graph $G_t = (\mathcal{U}, E_t)$ over users, which is initialized to be a complete graph (Algo.1 Line 2). Users with similar learned preferences will be connected with edges in $E_t$. The connected components in the graph represent the inferred clusters by the algorithm. At round $t$, user $i_t$ comes to be served with a feasible arm set $\mathcal{A}_t$ for the agent to choose from (Line 4). In Line 5, RCLUB-WCU detects the connected component $V_t$ in the graph containing user $i_t$ to be the current inferred cluster for $i_t$.

**Robust preference estimation of cluster $V_t$.** After determining the cluster $V_t$, RCLUB-WCU estimates the common preferences for users in $V_t$ using the historical feedback of all users in $V_t$ and recommends an arm accordingly. The corrupted behaviors could cause inaccurate preference estimates, which can easily mislead the agent. To address this, inspired by [40, 12], we use weighted ridge regression to make corruption-robust estimations. Specifically, RCLUB-WCU robustly estimates the common preference vector of cluster $V_t$ by solving the following weighted ridge regression

$$\hat{\boldsymbol{\theta}}_{V_t,t-1} = \underset{\boldsymbol{\theta} \in \mathbb{R}^d}{\arg\min} \sum_{\substack{s \in [t-1] \\ i_s \in V_t}} w_{i_s,s}(r_s - \boldsymbol{x}_{a_s}^\top \boldsymbol{\theta})^2 + \lambda \|\boldsymbol{\theta}\|_2^2 , \tag{2}$$

where $\lambda > 0$ is a regularization coefficient. Its closed-form solution is $\hat{\boldsymbol{\theta}}_{V_t,t-1} = \boldsymbol{M}_{V_t,t-1}^{-1} \boldsymbol{b}_{V_t,t-1} ,$ where $\boldsymbol{M}_{V_t,t-1} = \lambda \boldsymbol{I} + \sum_{\substack{s \in [t-1] \\ i_s \in V_t}} w_{i_s,s} \boldsymbol{x}_{a_s} \boldsymbol{x}_{a_s}^\top, \boldsymbol{b}_{V_t,t-1} = \sum_{\substack{s \in [t-1] \\ i_s \in V_t}} w_{i_s,s} r_{a_s} \boldsymbol{x}_{a_s} .$

We set the weight of sample for user $i_s$ in $V_t$ at round $s$ as $w_{i_s,s} = \min\{1, \alpha/\|\boldsymbol{x}_{a_s}\|_{M'^{-1}_{i_s,s}}\}$, where $\alpha$ is a coefficient to be determined later. The intuitions of designing these weights are as follows. The term $\|\boldsymbol{x}_{a_s}\|_{M'^{-1}_{i_s,s}}$ is the confidence radius of arm $a_s$ for user $i_s$ at $s$, reflecting how confident the algorithm is about the estimation of $i_s$'s preference on $a_s$ at $s$. If $\|\boldsymbol{x}_{a_s}\|_{M'^{-1}_{i_s,s}}$ is large, it means the agent is uncertain of user $i_s$'s preference on $a_s$, indicating this sample is probably corrupted.

Therefore, we use the inverse of confidence radius to assign a small weight to this round's sample if it is potentially corrupted. In this way, uncertain information for users in cluster $V_t$ is assigned with less importance when estimating the $V_t$'s preference vector, which could help relieve the estimation inaccuracy caused by corruption. For technical details, please refer to Section 5.1 and Appendix.

**Recommend $a_t$ with estimated preference of cluster $V_t$.** Based on the corruption-robust preference estimation $\hat{\boldsymbol{\theta}}_{V_t,t-1}$ of cluster $V_t$, in Line 7, the agent recommends an arm using the upper confidence bound (UCB) strategy to balance exploration and exploitation

$$a_t = \operatorname{argmax}_{a \in \mathcal{A}_t} \boldsymbol{x}_a^\top \hat{\boldsymbol{\theta}}_{V_t,t-1} + \beta \|\boldsymbol{x}_a\|_{\boldsymbol{M}_{V_t,t-1}^{-1}} \triangleq \hat{R}_{a,t} + C_{a,t} , \tag{3}$$

where $\beta = \sqrt{\lambda} + \sqrt{2 \log(\frac{1}{\delta}) + d \log(1 + \frac{T}{\lambda d})} + \alpha C$ is the confidence radius parameter, $\hat{R}_{a,t}$ denotes the estimated reward of arm $a$ at $t$, $C_{a,t}$ denotes the confidence radius of arm $a$ at $t$. The design of $C_{a,t}$ theoretically relies on Lemma 2 that will be given in Section 5.

**Update the robust estimation of user $i_t$.** After receiving $r_t$, the algorithm updates the estimation statistics of user $i_t$, while keeping the statistics of others unchanged (Line 8 and Line 9). Specifically, RCLUB-WCU estimates the preference vector of user $i_t$ by solving a weighted ridge regression

$$\hat{\boldsymbol{\theta}}_{i_t,t} = \underset{\boldsymbol{\theta} \in \mathbb{R}^d}{\arg\min} \sum_{\substack{s \in [t] \\ i_s = i_t}} w_{i_s,s}(r_s - \boldsymbol{x}_{a_s}^\top \boldsymbol{\theta})^2 + \lambda \|\boldsymbol{\theta}\|_2^2 \tag{4}$$

with closed-form solution $\hat{\boldsymbol{\theta}}_{i_t,t} = (\lambda \boldsymbol{I} + \boldsymbol{M}_{i_t,t})^{-1} \boldsymbol{b}_{i_t,t}$, where $\boldsymbol{M}_{i_t,t} = \sum_{\substack{s \in [t] \\ i_s = i_t}} w_{i_s,s} \boldsymbol{x}_{a_s} \boldsymbol{x}_{a_s}^\top$, $\boldsymbol{b}_{i_t,t} = \sum_{\substack{s \in [t] \\ i_s = i_t}} w_{i_s,s} r_{a_s} \boldsymbol{x}_{a_s}$, and we design the weights in the same way by the same reasoning.

**Update the dynamic graph.** Finally, with the updated statistics of user $i_t$, RCLUB-WCU checks

---

**Algorithm 2** OCCUD (At round $t$, used in Line 11 in Algo.1)

1: Initialize $\tilde{\mathcal{U}}_t = \emptyset$; input probability parameter $\delta$.
2: Update the statistics for non-robust estimation of user $i_t$
  $\tilde{\boldsymbol{M}}_{i_t,t} = \tilde{\boldsymbol{M}}_{i_t,t-1} + \boldsymbol{x}_{a_t} \boldsymbol{x}_{a_t}^\top , \tilde{\boldsymbol{b}}_{i_t,t} = \tilde{\boldsymbol{b}}_{i_t,t-1} + r_t \boldsymbol{x}_{a_t} , \tilde{\boldsymbol{\theta}}_{i_t,t} = (\lambda \boldsymbol{I} + \tilde{\boldsymbol{M}}_{i_t,t})^{-1} \tilde{\boldsymbol{b}}_{i_t,t} ,$
3: Keep non-robust estimation statistics of other users unchanged
  $\tilde{\boldsymbol{M}}_{\ell,t} = \tilde{\boldsymbol{M}}_{\ell,t-1}, \tilde{\boldsymbol{b}}_{\ell,t} = \tilde{\boldsymbol{b}}_{\ell,t-1}, \tilde{\boldsymbol{\theta}}_{\ell,t} = \tilde{\boldsymbol{\theta}}_{\ell,t-1},$ for all $\ell \in \mathcal{U}, \ell \neq i_t .$
4: **for all** connected component $V_{j,t} \in G_t$ **do**
5:   Calculate the robust estimation statistics for the cluster $V_{j,t}$:
    $\boldsymbol{M}_{V_{j,t},t} = \lambda \boldsymbol{I} + \sum_{\ell \in V_{j,t}} \boldsymbol{M}_{\ell,t} , T_{V_{j,t},t} = \sum_{\ell \in V_{j,t}} T_{\ell,t} ,$
    $\boldsymbol{b}_{V_{j,t},t} = \sum_{\ell \in V_{j,t}} \boldsymbol{b}_{\ell,t} , \hat{\boldsymbol{\theta}}_{V_{j,t},t} = \boldsymbol{M}_{V_{j,t},t}^{-1} \boldsymbol{b}_{V_{j,t},t} ;$
6:   **for all** user $i \in V_{j,t}$ **do**
7:     Detect user $i$ to be a corrupted user and add user $i$ to the set $\tilde{\mathcal{U}}_t$ if the following holds:

$$\left\| \tilde{\boldsymbol{\theta}}_{i,t} - \hat{\boldsymbol{\theta}}_{V_{i,t},t} \right\|_2 > \frac{\sqrt{d \log(1 + \frac{T_{i,t}}{\lambda d}) + 2 \log(\frac{1}{\delta})} + \sqrt{\lambda}}{\sqrt{\lambda_{\min}(\tilde{\boldsymbol{M}}_{i,t}) + \lambda}} + \frac{\sqrt{d \log(1 + \frac{T_{V_{i,t},t}}{\lambda d}) + 2 \log(\frac{1}{\delta})} + \sqrt{\lambda} + \alpha C}{\sqrt{\lambda_{\min}(\boldsymbol{M}_{V_{i,t},t})}} ,$$
$$\tag{5}$$

    where $\lambda_{\min}(\cdot)$ denotes the minimum eigenvalue of the matrix argument.
8:   **end for**
9: **end for**

---

whether the inferred $i_t$'s preference similarities with other users are still true, and updates the graph accordingly. Precisely, if gap between the updated estimation $\hat{\boldsymbol{\theta}}_{i_t,t}$ of $i_t$ and the estimation $\hat{\boldsymbol{\theta}}_{\ell,t}$ of user $\ell$ exceeds a threshold in Line 10, RCLUB-WCU will delete the edge $(i_t, \ell)$ in $G_{t-1}$ to split them apart. The threshold is carefully designed to handle the estimation uncertainty from both stochastic noises and potential corruptions. The updated graph $G_t = (\mathcal{U}, E_t)$ will be used in the next round.

## 4.2 OCCUD

Based on the inferred clustering structure of RCLUB-WCU, we devise a novel online detection algorithm OCCUD (Algo.2). The design ideas and process of OCCUD are as follows.

Besides the robust preference estimations (with weighted regression) of users and clusters kept by RCLUB-WCU, OCCUD also maintains the non-robust estimations for each user by online ridge regression without weights (Line 2 and Line 3). Specifically, at round $t$, OCCUD updates the non-robust estimation of user $i_t$ by solving the following online ridge regression:

$$\tilde{\boldsymbol{\theta}}_{i_t,t} = \underset{\boldsymbol{\theta} \in \mathbb{R}^d}{\arg\min} \sum_{\substack{s \in [t] \\ i_s = i_t}} (r_s - \boldsymbol{x}_{a_s}^\top \boldsymbol{\theta})^2 + \lambda \|\boldsymbol{\theta}\|_2^2 \,, \tag{6}$$

with solution $\tilde{\boldsymbol{\theta}}_{i_t,t} = (\lambda \boldsymbol{I} + \tilde{\boldsymbol{M}}_{i_t,t})^{-1} \tilde{\boldsymbol{b}}_{i_t,t}$, where $\tilde{\boldsymbol{M}}_{i_t,t} = \sum_{\substack{s \in [t] \\ i_s = i_t}} \boldsymbol{x}_{a_s} \boldsymbol{x}_{a_s}^\top$, $\tilde{\boldsymbol{b}}_{i_t,t} = \sum_{\substack{s \in [t] \\ i_s = i_t}} r_{a_s} \boldsymbol{x}_{a_s}$.

With the robust and non-robust preference estimations, OCCUD does the following to detect corrupted users based on the clustering structure inferred by RCLUB-WCU. First, OCCUD finds the connected components in the graph kept by RCLUB-WCU, which represent the inferred clusters. Then, for each inferred cluster $V_{j,t} \in G_t$: (1) OCCUD computes its robustly estimated preferences vector $\hat{\boldsymbol{\theta}}_{V_{j,t},t}$ (Line 5). (2) For each user $i$ whose inferred cluster is $V_{j,t}$ (*i.e.*, $i \in V_{j,t}$), OCCUD computes the gap between user $i$'s non-robustly estimated preference vector $\tilde{\boldsymbol{\theta}}_{i,t}$ and the robust estimation $\hat{\boldsymbol{\theta}}_{V_{i,t},t}$ for user $i$'s inferred cluster $V_{j,t}$. If the gap exceeds a carefully-designed threshold, OCCUD will detect user $i$ as corrupted and add $i$ to the detected corrupted user set $\tilde{\mathcal{U}}_t$ (Line 7).

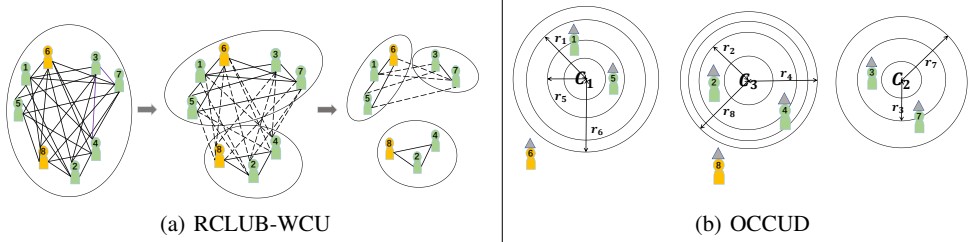

(a) RCLUB-WCU        (b) OCCUD

Figure 2: Algorithm illustrations. Users 6 and 8 are corrupted users (orange), and the others are normal (green). (a) illustrates RCLUB-WCU, which starts with a complete user graph, and adaptively deletes edges between users (dashed lines) with dissimilar robustly learned preferences. The corrupted behaviors of users 6 and 8 may cause inaccurate preference estimations, leading to erroneous relation inference. In this case, how to delete edges correctly is non-trivial, and RCLUB-WCU addresses this challenge (detailed in Section 4.1). (b) illustrates OCCUD at some round $t$, where person icons with triangle hats represent the non-robust user preference estimations. The gap between the non-robust estimation of user 6 and the robust estimation of user 6's inferred cluster (circle $C_1$) exceeds the threshold $r_6$ at this round (Line 7 in Algo.2), so OCCUD detects user 6 to be corrupted.

The intuitions of OCCUD are as follows. On the one hand, after some interactions, RCLUB-WCU will infer the user clustering structure accurately. Thus, at round $t$, the robust estimation $\hat{\boldsymbol{\theta}}_{V_{i,t},t}$ for user $i$'s inferred cluster should be pretty close to user $i$'s ground-truth preference vector $\boldsymbol{\theta}_i$. On the other hand, since the feedback of normal users are always regular, at round $t$, if user $i$ is a normal user, the non-robust estimation $\tilde{\boldsymbol{\theta}}_{i,t}$ should also be close to the ground-truth $\boldsymbol{\theta}_i$. However, the non-robust estimation of a corrupted user should be quite far from the ground truth due to corruptions. Based on this reasoning, OCCUD compares each user's non-robust estimation and the robust estimation of the user's inferred cluster to detect the corrupted users. For technical details, please refer to Section 5.2 and Appendix. Simple illustrations of our proposed algorithms can be found in Fig.2.

## 5 Theoretical Analysis

In this section, we theoretically analyze the performances of our proposed algorithms, RCLUB-WCU and OCCUD. Due to the page limit, we put the proofs in the Appendix.

### 5.1 Regret Analysis of RCLUB-WCU

This section gives an upper bound of the expected regret (defined in Eq.(1)) for RCLUB-WCU.

The following lemma provides a sufficient time $T_0(\delta)$, after which RCLUB-WCU can cluster all the users correctly with high probability.

**Lemma 1.** *With probability at least $1 - 3\delta$, RCLUB-WCU will cluster all the users correctly after*

$$T_0(\delta) \triangleq 16u \log(\frac{u}{\delta}) + 4u \max\{\frac{288d}{\gamma^2 \alpha \sqrt{\lambda}\tilde{\lambda}_x} \log(\frac{u}{\delta}), \frac{16}{\tilde{\lambda}_x^2} \log(\frac{8d}{\tilde{\lambda}_x^2 \delta}), \frac{72\sqrt{\lambda}}{\alpha\gamma^2 \tilde{\lambda}_x}, \frac{72\alpha C^2}{\gamma^2 \sqrt{\lambda}\tilde{\lambda}_x}\}$$

*for some $\delta \in (0, \frac{1}{3})$, where $\tilde{\lambda}_x \triangleq \int_0^{\lambda_x} (1 - e^{-\frac{(\lambda_x - x)^2}{2\sigma^2}})^K dx$, $|\mathcal{A}_t| \leq K, \forall t \in [T]$.*

After $T_0(\delta)$, the following lemma gives a bound of the gap between $\hat{\boldsymbol{\theta}}_{V_t,t-1}$ and the ground-truth $\boldsymbol{\theta}_{i_t}$ in direction of action vector $\boldsymbol{x}_a$ for RCLUB-WCU, which supports the design in Eq.(3).

**Lemma 2.** *With probability at least $1 - 4\delta$ for some $\delta \in (0, \frac{1}{4})$, $\forall t \geq T_0(\delta)$, we have:*

$$\left| \boldsymbol{x}_a^{\mathrm{T}} (\hat{\boldsymbol{\theta}}_{V_t,t-1} - \boldsymbol{\theta}_{i_t}) \right| \leq \beta \|\boldsymbol{x_a}\|_{M_{V_t,t-1}^{-1}} \triangleq C_{a,t} .$$

With Lemma 1 and 2, we prove the following theorem on the regret upper bound of RCLUB-WCU.

**Theorem 3** (**Regret Upper Bound of RCLUB-WCU**). *With the assumptions in Section 3, and picking $\alpha = \frac{\sqrt{d}+\sqrt{\lambda}}{C}$, the expected regret of the RCLUB-WCU algorithm for $T$ rounds satisfies*

$$R(T) \leq O\big((\frac{C\sqrt{d}}{\gamma^2 \tilde{\lambda}_x} + \frac{1}{\tilde{\lambda}_x^2})u \log(T)\big) + O\big(d\sqrt{mT}\log(T)\big) + O\big(mCd\log^{1.5}(T)\big) . \tag{7}$$

**Discussion and Comparison.** The regret bound in Eq.(7) has three terms. The first term is the time needed to get enough information for accurate robust estimations such that RCLUB-WCU could cluster all users correctly afterward with high probability. This term is related to the *corruption level $C$*, which is inevitable since, if there are more corrupted user feedback, it will be harder for the algorithm to learn the clustering structure correctly. The last two terms correspond to the regret after $T_0$ with the correct clustering. Specifically, the second term is caused by stochastic noises when leveraging the aggregated information within clusters to make recommendations; the third term associated with the *corruption level $C$* is the regret caused by the disruption of corrupted behaviors.

When the *corruption level $C$* is *unknown*, we can use its estimated upper bound $\hat{C} \triangleq \sqrt{T}$ to replace $C$ in the algorithm. In this way, if $C \leq \hat{C}$, the bound will be replacing $C$ with $\hat{C}$ in Eq.(7); when $C > \sqrt{T}$, $R(T) = O(T)$, which is already optimal for a large class of bandit algorithms [12].

The following theorem gives a regret lower bound of the LOCUD problem.

**Theorem 4** (Regret lower bound for LOCUD). *There exists a problem instance for the LOCUD problem such that for any algorithm*

$$R(T) \geq \Omega(d\sqrt{mT} + dC) .$$

Its proof and discussions can be found in Appendix D. The upper bound in Theorem 3 asymptotically matches this lower bound in $T$ up to logarithmic factors, showing our regret bound is nearly optimal.

We then compare our regret upper bound with several degenerated cases. First, when $C = 0$, *i.e.*, all users are normal, our setting degenerates to the classic CB problem [8]. In this case the bound in Theorem 3 becomes $O(1/\tilde{\lambda}_x^2 \cdot u \log(T)) + O(d\sqrt{mT}\log(T))$, perfectly matching the state-of-the-art results in CB [8, 19, 21]. Second, when $m = 1$ and $u = 1$, *i.e.*, there is only one user, our setting degenerates to linear bandits with adversarial corruptions [22, 12], and the bound in Theorem 3 becomes $O(d\sqrt{T}\log(T)) + O(Cd\log^{1.5}(T))$, it also perfectly matches the nearly optimal result in [12]. The above comparisons also show the tightness of the regret bound of RCLUB-WCU.

## 5.2 Theoretical Performance Guarantee for OCCUD

The following theorem gives a performance guarantee of the online detection algorithm OCCUD.

**Theorem 5** (**Theoretical Guarantee for OCCUD**). *With assumptions in Section 3, at $\forall t \geq T_0(\delta)$, for any detected corrupted user $i \in \tilde{\mathcal{U}}_t$, with probability at least $1 - 5\delta$, $i$ is indeed a corrupted user.*

This theorem guarantees that after RCLUB-WCU learns the clustering structure accurately, with high probability, the corrupted users detected by OCCUD are indeed corrupted, showing the high detection accuracy of OCCUD. The proof of Theorem 5 can be found in Appendix D.

# 6 Experiments

This section shows experimental results on synthetic and real data to evaluate RCLUB-WCU's recommendation quality and OCCUD's detection accuracy. We compare RCLUB-WCU to LinUCB [1] with a single non-robust estimated vector for all users, LinUCB-Ind with separate non-robust estimated vectors for each user, CW-OFUL [12] with a single robust estimated vector for all users, CW-OFUL-Ind with separate robust estimated vectors for each user, CLUB[8], and SCLUB[21]. More description of these baselines are in Appendix F. To show that the design of OCCUD is nontrivial, we develop a straightforward detection algorithm GCUD, which leverages the same cluster structure as OCCUD but detects corrupted users by selecting users with highest $\left\| \hat{\boldsymbol{\theta}}_{i,t} - \hat{\boldsymbol{\theta}}_{V_{i,t},t-1} \right\|_2$ in each inferred cluster. GCUD selects users according to the underlying percentage of corrupted users, which is unrealistic in practice, but OCCUD still performs better in this unfair condition.

**Remark.** The offline detection methods [39, 6, 18, 32] need to know all the user information in advance to derive the user embedding for classification, so they cannot be directly applied in online detection with bandit feedback thus cannot be directly compared to OCCUD. However, we observe the AUC achieved by OCCUD on Amazon and Yelp (in Tab.1) is similar to recent offline methods [18, 32]. Additionally, OCCUD has rigorous theoretical performance guarantee (Section 5.2).

## 6.1 Experiments on Synthetic Dataset

We use $u = 1,000$ users and $m = 10$ clusters, where each cluster contains 100 users. We randomly select 100 users as the corrupted users. The preference and arm (item) vectors are drawn in $d - 1$ ($d = 50$) dimensions with each entry a standard Gaussian variable and then normalized, added one more dimension with constant 1, and divided by $\sqrt{2}$ [21]. We fix an arm set with $|\mathcal{A}| = 1000$ items, at each round, 20 items are randomly selected to form a set $\mathcal{A}_t$ to choose from. Following [40, 3], in the first $k$ rounds, we always flip the reward of corrupted users by setting $r_t = -\boldsymbol{x}_{a_t}^{\mathrm{T}} \boldsymbol{\theta}_{i_t,t} + \eta_t$. And we leave the remaining $T - k$ rounds intact. Here we set $T = 1,000,000$ and $k = 20,000$.

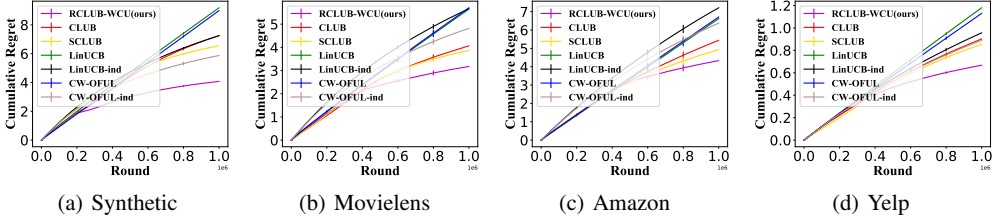

|     |     |     |     |
| --- | --- | --- | --- |
| (a) Synthetic | (b) Movielens | (c) Amazon | (d) Yelp |

Figure 3: Recommendation results on the synthetic and real-world datasets

Fig.3(a) shows the recommendation results. RCLUB-WCU outperforms all baselines and achieves a sub-linear regret. LinUCB and CW-OFUL perform worst as they ignore the preference differences among users. CW-OFUL-Ind outperforms LinUCB-Ind because it considers the corruption, but worse than RCLUB-WCU since it does not consider leveraging user relations to speed up learning.

The detection results are shown in Tab.1. We test the AUC of OCCUD and GCUD in every $200,000$ rounds. OCCUD's performance improves over time with more interactions, while GCUD's performance is much worse as it detects corrupted users only relying on the robust estimations. OCCUD finally achieves an AUC of 0.855, indicating it can identify most of the corrupted users.

## 6.2 Experiments on Real-world Datasets

We use three real-world data Movielens [11], Amazon[31], and Yelp [33]. The Movielens data does not have the corrupted users' labels, so following [24], we manually select the corrupted users. On Amazon data, following [39], we label the users with more than 80% helpful votes as normal users, and label users with less than 20% helpful votes as corrupted users. The Yelp data contains users and their comments on restaurants with true labels of the normal users and corrupted users.

We select 1,000 users and 1,000 items for Movielens; 1,400 users and 800 items for Amazon; 2,000 users and 2,000 items for Yelp. The ratios of corrupted users on these data are 10%, 3.5%, and

30.9%, respectively. We generate the preference and item vectors following [37, 21]. We first construct the binary feedback matrix through the users' ratings: if the rating is greater than 3, the feedback is 1; otherwise, the feedback is 0. Then we use SVD to decompose the extracted binary feedback matrix $R_{u \times m} = \boldsymbol{\theta} S X^{\mathrm{T}}$, where $\boldsymbol{\theta} = (\boldsymbol{\theta}_i), i \in [u]$ and $X = (\boldsymbol{x}_j), j \in [m]$, and select $d = 50$ dimensions. We have 10 clusters on Movielens and Amazon, and 20 clusters on Yelp. We use the same corruption mechanism as the synthetic data with $T = 1,000,000$ and $k = 20,000$. We conduct more experiments in different environments to show our algorithms' robustness in Appendix.G.

The recommendation results are shown in Fig.3(b)-(d). RCLUB-WCU outperforms all baselines. On the Amazon dataset, the percentage of corrupted users is lowest, RCLUB-WCU's advantages over baselines decrease because of the weakened corruption. The corrupted user detection results are provided in Tab.1. OCCUD's performance improves over time and is much better than GCUD. On the Movielens dataset, OCCUD achieves an AUC of 0.85; on the Amazon dataset, OCCUD

| Dataset | Time Alg | 0.2M | 0.4M | 0.6M | 0.8M | 1M |
|---|---|---|---|---|---|---|
| Synthetic | OCCUD | 0.599 | 0.651 | 0.777 | 0.812 | **0.855** |
| | GCUD | 0.477 | 0.478 | 0.483 | 0.484 | 0.502 |
| Movielens | OCCUD | 0.65 | 0.750 | 0.785 | 0.83 | **0.85** |
| | GCUD | 0.450 | 0.474 | 0.485 | 0.489 | 0.492 |
| Amazon | OCCUD | 0.639 | 0.735 | 0.761 | 0.802 | **0.840** |
| | GCUD | 0.480 | 0.480 | 0.486 | 0.500 | 0.518 |
| Yelp | OCCUD | 0.452 | 0.489 | 0.502 | 0.578 | **0.628** |
| | GCUD | 0.473 | 0.481 | 0.496 | 0.500 | 0.510 |

Table 1: Detection results on synthetic and real datasets

achieves an AUC of 0.84; and on the Yelp dataset, OCCUD achieves an AUC of 0.628. According to recent works on offline settings [18, 32], our results are relatively high.

## 7 Conclusion

In this paper, we are the first to propose the novel LOCUD problem, where there are many users with *unknown* preferences and *unknown* relations, and some corrupted users can occasionally perform disrupted actions to fool the agent. Hence, the agent not only needs to learn the *unknown* user preferences and relations robustly from potentially disrupted bandit feedback, balance the exploration-exploitation trade-off to minimize regret, but also needs to detect the corrupted users over time. To robustly learn and leverage the *unknown* user preferences and relations from corrupted behaviors, we propose a novel bandit algorithm RCLUB-WCU. To detect the corrupted users in the online bandit setting, based on the learned user relations of RCLUB-WCU, we propose a novel detection algorithm OCCUD. We prove a regret upper bound for RCLUB-WCU, which matches the lower bound asymptotically in $T$ up to logarithmic factors and matches the state-of-the-art results in degenerate cases. We also give a theoretical guarantee for the detection accuracy of OCCUD. Extensive experiments show that our proposed algorithms achieve superior performance over previous bandit algorithms and high corrupted user detection accuracy.

## 8 Acknowledgement

The corresponding author Shuai Li is supported by National Key Research and Development Program of China (2022ZD0114804) and National Natural Science Foundation of China (62376154, 62006151, 62076161). The work of John C.S. Lui was supported in part by the RGC's SRFS2122-4S02.

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

# Appendix

## A  Proof of Lemma 1

We first prove the following result:
With probability at least $1 - \delta$ for some $\delta \in (0, 1)$, at any $t \in [T]$:

$$\left\| \hat{\boldsymbol{\theta}}_{i,t} - \boldsymbol{\theta}^{j(i)} \right\|_2 \leq \frac{\beta(T_{i,t}, \frac{\delta}{u})}{\sqrt{\lambda + \lambda_{\min}(\boldsymbol{M}_{i,t})}}, \forall i \in \mathcal{U}, \tag{8}$$

where $\beta(T_{i,t}, \frac{\delta}{u}) \triangleq \sqrt{2 \log(\frac{u}{\delta}) + d \log(1 + \frac{T_{i,t}}{\lambda d})} + \sqrt{\lambda} + \alpha C$.

$$\begin{aligned}
\hat{\boldsymbol{\theta}}_{i,t} - \boldsymbol{\theta}^{j(i)} &= (\lambda \boldsymbol{I} + \boldsymbol{M}_{i,t})^{-1} \boldsymbol{b}_{i,t} - \boldsymbol{\theta}^{j(i)} \\
&= (\lambda \boldsymbol{I} + \sum_{\substack{s \in [t] \\ i_s = i}} w_{i_s,s} \boldsymbol{x}_{a_s} \boldsymbol{x}_{a_s}^\top)^{-1} \sum_{\substack{s \in [t] \\ i_s = i}} w_{i_s,s} \boldsymbol{x}_{a_s} r_s - \boldsymbol{\theta}^{j(i)} \\
&= (\lambda \boldsymbol{I} + \sum_{\substack{s \in [t] \\ i_s = i}} w_{i_s,s} \boldsymbol{x}_{a_s} \boldsymbol{x}_{a_s}^\top)^{-1} \left( \sum_{\substack{s \in [t] \\ i_s = i}} w_{i_s,s} \boldsymbol{x}_{a_s} (\boldsymbol{x}_{a_s}^\top \boldsymbol{\theta}_{i_s} + \eta_s + c_s) \right) - \boldsymbol{\theta}^{j(i)} \\
&= (\lambda \boldsymbol{I} + \sum_{\substack{s \in [t] \\ i_s = i}} w_{i_s,s} \boldsymbol{x}_{a_s} \boldsymbol{x}_{a_s}^\top)^{-1} \Bigg[ (\lambda \boldsymbol{I} + \sum_{\substack{s \in [t] \\ i_s = i}} w_{i_s,s} \boldsymbol{x}_{a_s} \boldsymbol{x}_{a_s}^\top) \boldsymbol{\theta}^{j(i)} - \lambda \boldsymbol{\theta}^{j(i)} + \sum_{\substack{s \in [t] \\ i_s = i}} w_{i_s,s} \boldsymbol{x}_{a_s} \eta_s \\
&\qquad + \sum_{\substack{s \in [t] \\ i_s = i}} w_{i_s,s} \boldsymbol{x}_{a_s} c_s \Bigg] - \boldsymbol{\theta}^{j(i)} \\
&= -\lambda \boldsymbol{M}_{i,t}'^{-1} \boldsymbol{\theta}^{j(i)} + \boldsymbol{M}_{i,t}'^{-1} \sum_{\substack{s \in [t] \\ i_s = i}} w_{i_s,s} \boldsymbol{x}_{a_s} \eta_s + \boldsymbol{M}_{i,t}'^{-1} \sum_{\substack{s \in [t] \\ i_s = i}} w_{i_s,s} \boldsymbol{x}_{a_s} c_s,
\end{aligned}$$

where we denote $\boldsymbol{M}_{i,t}' = \boldsymbol{M}_{i,t} + \lambda \boldsymbol{I}$, and the above equations hold by definition.

Therefore, we have

$$\left\| \hat{\boldsymbol{\theta}}_{i,t} - \boldsymbol{\theta}^{j(i)} \right\|_2 \leq \lambda \left\| \boldsymbol{M}_{i,t}'^{-1} \boldsymbol{\theta}^{j(i)} \right\|_2 + \left\| \boldsymbol{M}_{i,t}'^{-1} \sum_{\substack{s \in [t] \\ i_s = i}} w_{i_s,s} \boldsymbol{x}_{a_s} \eta_s \right\|_2 + \left\| \boldsymbol{M}_{i,t}'^{-1} \sum_{\substack{s \in [t] \\ i_s = i}} w_{i_s,s} \boldsymbol{x}_{a_s} c_s \right\|_2. \tag{9}$$

We then bound the three terms in Eq.(9) one by one. For the first term:

$$\lambda \left\| \boldsymbol{M}_{i,t}'^{-1} \boldsymbol{\theta}^{j(i)} \right\|_2 \leq \lambda \left\| \boldsymbol{M}_{i,t}'^{-\frac{1}{2}} \right\|_2^2 \left\| \boldsymbol{\theta}^{j(i)} \right\|_2 \leq \frac{\sqrt{\lambda}}{\sqrt{\lambda_{\min}(\boldsymbol{M}_{i,t}')}}, \tag{10}$$

where we use the Cauchy–Schwarz inequality, the inequality for the operator norm of matrices, and the fact that $\lambda_{\min}(\boldsymbol{M}_{i,t}') \geq \lambda$.

For the second term in Eq.(9), we have

$$\left\| \boldsymbol{M}_{i,t}'^{-1} \sum_{\substack{s \in [t] \\ i_s = i}} w_{i_s,s} \boldsymbol{x}_{a_s} \eta_s \right\|_2 \leq \left\| \boldsymbol{M}_{i,t}'^{-\frac{1}{2}} \sum_{\substack{s \in [t] \\ i_s = i}} w_{i_s,s} \boldsymbol{x}_{a_s} \eta_s \right\|_2 \left\| \boldsymbol{M}_{i,t}'^{-\frac{1}{2}} \right\|_2 \tag{11}$$

$$= \frac{\left\| \sum_{\substack{s \in [t] \\ i_s = i}} w_{i_s,s} \boldsymbol{x}_{a_s} \eta_s \right\|_{\boldsymbol{M}_{i,t}'^{-1}}}{\sqrt{\lambda_{\min}(\boldsymbol{M}_{i,t}')}}, \tag{12}$$

where Eq.(11) follows by the Cauchy–Schwarz inequality and the inequality for the operator norm of matrices, and Eq.(12) follows by the Courant-Fischer theorem.

Let $\tilde{\boldsymbol{x}}_s \triangleq \sqrt{w_{i_s,s}}\boldsymbol{x}_{a_s}$, $\tilde{\eta}_s \triangleq \sqrt{w_{i_s,s}}\eta_s$, then we have: $\|\tilde{\boldsymbol{x}}_s\|_2 \leq \|\sqrt{w_{i_s,s}}\|_2 \|\boldsymbol{x}_{a_s}\|_2 \leq 1$, $\tilde{\eta}_s$ is still 1-sub-gaussian (since $\eta_s$ is 1-sub-gaussian and $\sqrt{w_{i_s,s}} \leq 1$), $\boldsymbol{M}'_{i,t} = \lambda \boldsymbol{I} + \sum_{\substack{s \in [t] \\ i_s=i}} \tilde{\boldsymbol{x}}_s \tilde{\boldsymbol{x}}_s^\top$, and the nominator in Eq.(12) becomes $\left\|\sum_{\substack{s \in [t] \\ i_s=i}} \tilde{\boldsymbol{x}}_s \tilde{\eta}_s\right\|_{\boldsymbol{M}'^{-1}_{i,t}}$. Then, following Theorem 1 in [1] and by union bound, with probability at least $1 - \delta$ for some $\delta \in (0,1)$, for any $i \in \mathcal{U}$, we have:

$$
\left\|\sum_{\substack{s \in [t] \\ i_s=i}} w_{i_s,s}\boldsymbol{x}_{a_s}\eta_s\right\|_{\boldsymbol{M}'^{-1}_{i,t}} = \left\|\sum_{\substack{s \in [t] \\ i_s=i}} \tilde{\boldsymbol{x}}_s \tilde{\eta}_s\right\|_{\boldsymbol{M}'^{-1}_{i,t}}
$$

$$
\leq \sqrt{2\log(\frac{u}{\delta}) + \log(\frac{\det(\boldsymbol{M}'_{i,t})}{\det(\lambda \boldsymbol{I})})}
$$

$$
\leq \sqrt{2\log(\frac{u}{\delta}) + d\log(1 + \frac{T_{i,t}}{\lambda d})}, \tag{13}
$$

where $\det(\cdot)$ denotes the determinant of matrix arguement, Eq.(13) is because $\det(\boldsymbol{M}'_{i,t}) \leq \left(\frac{\text{trace}(\lambda \boldsymbol{I} + \sum_{\substack{s \in [t] \\ i_s=i}} w_{i_s,s}\boldsymbol{x}_{a_s}\boldsymbol{x}_{a_s}^\top)}{d}\right)^d \leq \left(\frac{\lambda d + T_{i,t}}{d}\right)^d$, and $\det(\lambda \boldsymbol{I}) = \lambda^d$.

For the third term in Eq.(9), we have

$$
\left\|\boldsymbol{M}'^{-1}_{i,t} \sum_{\substack{s \in [t] \\ i_s=i}} w_{i_s,s}\boldsymbol{x}_{a_s}c_s\right\|_2 \leq \left\|\boldsymbol{M}'^{-\frac{1}{2}}_{i,t} \sum_{\substack{s \in [t] \\ i_s=i}} w_{i_s,s}\boldsymbol{x}_{a_s}c_s\right\|_2 \left\|\boldsymbol{M}'^{-\frac{1}{2}}_{i,t}\right\|_2 \tag{14}
$$

$$
= \frac{\left\|\sum_{\substack{s \in [t] \\ i_s=i}} w_{i_s,s}\boldsymbol{x}_{a_s}c_s\right\|_{\boldsymbol{M}'^{-1}_{i,t}}}{\sqrt{\lambda_{\min}(\boldsymbol{M}'_{i,t})}} \tag{15}
$$

$$
\leq \frac{\sum_{\substack{s \in [t] \\ i_s=i}} |c_s| w_{i,s} \|\boldsymbol{x}_{a_s}\|_{\boldsymbol{M}'^{-1}_{i,t}}}{\sqrt{\lambda_{\min}(\boldsymbol{M}'_{i,t})}}
$$

$$
\leq \frac{\alpha C}{\sqrt{\lambda_{\min}(\boldsymbol{M}'_{i,t})}} \tag{16}
$$

where Eq.(14) follows by the Cauchy–Schwarz inequality and the inequality for the operator norm of matrices, Eq.(15) follows by the Courant-Fischer theorem, and Eq.(16) is because by definition $w_{i,s} \leq \frac{\alpha}{\|\boldsymbol{x}_{a_s}\|_{\boldsymbol{M}'^{-1}_{i,s}}} \leq \frac{\alpha}{\|\boldsymbol{x}_{a_s}\|_{\boldsymbol{M}'^{-1}_{i,t}}}$ (since $\boldsymbol{M}'_{i,t} \succeq \boldsymbol{M}'_{i,s}$, $\boldsymbol{M}'^{-1}_{i,s} \succeq \boldsymbol{M}'^{-1}_{i,t}$, $\|\boldsymbol{x}_{a_s}\|_{\boldsymbol{M}'^{-1}_{i,s}} \geq \|\boldsymbol{x}_{a_s}\|_{\boldsymbol{M}'^{-1}_{i,t}}$), $\sum_{t=1}^T |c_t| \leq C$.

Combining the above bounds of these three terms, we can get that Eq.(8) holds.

We then prove the following technical lemma.

**Lemma 6.** *Under Assumption 3, at any time $t$, for any fixed unit vector $\boldsymbol{\theta} \in \mathbb{R}^d$*

$$
\mathbb{E}_t[(\boldsymbol{\theta}^\top \boldsymbol{x}_{a_t})^2 \,|\, |\mathcal{A}_t|] \geq \tilde{\lambda}_x \triangleq \int_0^{\lambda_x} (1 - e^{-\frac{(\lambda_x - x)^2}{2\sigma^2}})^K dx, \tag{17}
$$

*where $K$ is the upper bound of $|\mathcal{A}_t|$ for any $t$.*

*Proof.* The proof of this lemma mainly follows the proof of Claim 1 in [8], but with more careful analysis, since their assumption on the arm generation distribution is more stringent than our Assumption 3 by putting more restrictions on the variance upper bound $\sigma^2$ (specifically, they require $\sigma^2 \leq \frac{\lambda^2}{8\log(4K)}$).

Denote the feasible arms at round $t$ by $\mathcal{A}_t = \{\boldsymbol{x}_{t,1}, \boldsymbol{x}_{t,2}, \ldots, \boldsymbol{x}_{t,|\mathcal{A}_t|}\}$. Consider the corresponding i.i.d. random variables $\theta_i = (\boldsymbol{\theta}^\top \boldsymbol{x}_{t,i})^2 - \mathbb{E}_t[(\boldsymbol{\theta}^\top \boldsymbol{x}_{t,i})^2 | |\mathcal{A}_t|], i = 1, 2, \ldots, |\mathcal{A}_t|$. By Assumption 3, $\theta_i$ s are sub-Gaussian random variables with variance bounded by $\sigma^2$. Therefore, for any $\alpha > 0$ and any $i \in [|\mathcal{A}_t|]$, we have:

$$\mathbb{P}_t(\theta_i < -\alpha | |\mathcal{A}_t|) \leq e^{-\frac{\alpha^2}{2\sigma^2}},$$

where we use $\mathbb{P}_t(\cdot)$ to be the shorthand for the conditional probability $\mathbb{P}(\cdot | (i_1, \mathcal{A}_1, r_1), \ldots, (i_{t-1}, \mathcal{A}_{t-1}, r_{t-1}), i_t)$.

By Assumption 3, we can also get that $\mathbb{E}_t[(\boldsymbol{\theta}^\top \boldsymbol{x}_{t,i})^2 | |\mathcal{A}_t| = \mathbb{E}_t[\boldsymbol{\theta}^\top \boldsymbol{x}_{t,i} \boldsymbol{x}_{t,i}^\top \boldsymbol{\theta} | |\mathcal{A}_t|] \geq \lambda_{\min}(\mathbb{E}_{\boldsymbol{x} \sim \rho}[\boldsymbol{x} \boldsymbol{x}^\top]) \geq \lambda_x$. With these inequalities above, we can get

$$\mathbb{P}_t(\min_{i=1,\ldots,|\mathcal{A}_t|} (\boldsymbol{\theta}^\top \boldsymbol{x}_{t,i})^2 \geq \lambda_x - \alpha | |\mathcal{A}_t|) \geq (1 - e^{-\frac{\alpha^2}{2\sigma^2}})^K.$$

Therefore, we can get

$$\mathbb{E}_t[(\boldsymbol{\theta}^\top \boldsymbol{x}_{a_t})^2 | |\mathcal{A}_t|] \geq \mathbb{E}_t[\min_{i=1,\ldots,|\mathcal{A}_t|} (\boldsymbol{\theta}^\top \boldsymbol{x}_{t,i})^2 | |\mathcal{A}_t|]$$

$$\geq \int_0^\infty \mathbb{P}_t(\min_{i=1,\ldots,|\mathcal{A}_t|} (\boldsymbol{\theta}^\top \boldsymbol{x}_{t,i})^2 \geq x | |\mathcal{A}_t|) dx$$

$$\geq \int_0^{\lambda_x} (1 - e^{-\frac{(\lambda_x - x)^2}{2\sigma^2}})^K dx \triangleq \tilde{\lambda}_x$$

$\square$

Note that $w_{i,s} = \min\{1, \frac{\alpha}{\|\boldsymbol{x}_{a_s}\|_{\boldsymbol{M}_{i,t}^{\prime -1}}}\}$, and we have

$$\frac{\alpha}{\|\boldsymbol{x}_{a_s}\|_{\boldsymbol{M}_{i,t}^{\prime -1}}} = \frac{\alpha}{\sqrt{\boldsymbol{x}_{a_s}^\top \boldsymbol{M}_{i,t}^{\prime -1} \boldsymbol{x}_{a_s}}} \geq \frac{\alpha}{\sqrt{\lambda_{\min}(\boldsymbol{M}_{i,t}^{\prime -1})}} = \alpha \sqrt{\lambda_{\min}(\boldsymbol{M}_{i,t}^\prime)} \geq \alpha \sqrt{\lambda}.$$

Since $\alpha \sqrt{\lambda} < 1$ typically holds, we have $w_{i,s} \geq \alpha \sqrt{\lambda}$.

Then, with the item regularity assumption stated in Assumption 3, the technical Lemma 6, together with Lemma 7 in [19], with probability at least $1 - \delta$, for a particular user $i$, at any $t$ such that $T_{i,t} \geq \frac{16}{\tilde{\lambda}_x^2} \log(\frac{8d}{\tilde{\lambda}_x^2 \delta})$, we have:

$$\lambda_{\min}(\boldsymbol{M}_{i,t}^\prime) \geq 2\alpha \sqrt{\lambda} \tilde{\lambda}_x T_{i,t} + \lambda. \tag{18}$$

With this result, together with Eq.(8), we can get that for any $t$ such that $T_{i,t} \geq \frac{16}{\tilde{\lambda}_x^2} \log(\frac{8d}{\tilde{\lambda}_x^2 \delta})$, with probability at least $1 - \delta$ for some $\delta \in (0, 1)$, $\forall i \in \mathcal{U}$, we have:

$$\left\| \hat{\boldsymbol{\theta}}_{i,t} - \boldsymbol{\theta}^{j(i)} \right\|_2 \leq \frac{\beta(T_{i,t}, \frac{\delta}{u})}{\sqrt{\lambda_{\min}(\boldsymbol{M}_{i,t}^\prime)}}$$

$$\leq \frac{\beta(T_{i,t}, \frac{\delta}{u})}{\sqrt{2\alpha \sqrt{\lambda} \tilde{\lambda}_x T_{i,t} + \lambda}}$$

$$\leq \frac{\beta(T_{i,t}, \frac{\delta}{u})}{\sqrt{2\alpha \sqrt{\lambda} \tilde{\lambda}_x T_{i,t}}}$$

$$= \frac{\sqrt{2 \log(\frac{u}{\delta}) + d \log(1 + \frac{T_{i,t}}{\lambda d})} + \sqrt{\lambda} + \alpha C}{\sqrt{2\alpha \sqrt{\lambda} \tilde{\lambda}_x T_{i,t}}}. \tag{19}$$

Then, we want to find a sufficient time $T_{i,t}$ for a fixed user $i$ such that

$$\left\| \hat{\boldsymbol{\theta}}_{i,t} - \boldsymbol{\theta}^{j(i)} \right\|_2 < \frac{\gamma}{4}. \tag{20}$$

To do this, with Eq.(19), we can get it by letting

$$\frac{\sqrt{\lambda}}{\sqrt{2\alpha\sqrt{\lambda}\tilde{\lambda}_x T_{i,t}}} < \frac{\gamma}{12}\,, \tag{21}$$

$$\frac{\alpha C}{\sqrt{2\alpha\sqrt{\lambda}\tilde{\lambda}_x T_{i,t}}} < \frac{\gamma}{12}\,, \tag{22}$$

$$\frac{\sqrt{2\log(\frac{u}{\delta}) + d\log(1 + \frac{T_{i,t}}{\lambda d})}}{\sqrt{2\alpha\sqrt{\lambda}\tilde{\lambda}_x T_{i,t}}} < \frac{\gamma}{12}\,. \tag{23}$$

For Eq.(21), we can get

$$T_{i,t} > \frac{72\sqrt{\lambda}}{\alpha\gamma^2\tilde{\lambda}_x}\,. \tag{24}$$

For Eq.(22), we can get

$$T_{i,t} > \frac{72\alpha C^2}{\gamma^2\sqrt{\lambda}\tilde{\lambda}_x}\,. \tag{25}$$

For Eq.(23), we have

$$\frac{2\log(\frac{u}{\delta}) + d\log(1 + \frac{T_{i,t}}{\lambda d})}{2\alpha\sqrt{\lambda}\tilde{\lambda}_x T_{i,t}} < \frac{\gamma^2}{144}\,. \tag{26}$$

Then it is sufficient to get Eq.(26) if the following holds

$$\frac{2\log(\frac{u}{\delta})}{2\alpha\sqrt{\lambda}\tilde{\lambda}_x T_{i,t}} < \frac{\gamma^2}{288}\,, \tag{27}$$

$$\frac{d\log(1 + \frac{T_{i,t}}{\lambda d})}{2\alpha\sqrt{\lambda}\tilde{\lambda}_x T_{i,t}} < \frac{\gamma^2}{288}\,. \tag{28}$$

For Eq.(27), we can get

$$T_{i,t} > \frac{288\log(\frac{u}{\delta})}{\gamma^2\alpha\sqrt{\lambda}\tilde{\lambda}_x} \tag{29}$$

For Eq.(28), we can get

$$T_{i,t} > \frac{144d}{\gamma^2\alpha\sqrt{\lambda}\tilde{\lambda}_x}\log(1 + \frac{T_{i,t}}{\lambda d})\,. \tag{30}$$

Following Lemma 9 in [19], we can get the following sufficient condition for Eq.(30):

$$T_{i,t} > \frac{288d}{\gamma^2\alpha\sqrt{\lambda}\tilde{\lambda}_x}\log(\frac{288}{\gamma^2\alpha\sqrt{\lambda}\tilde{\lambda}_x})\,. \tag{31}$$

Then, since typically $\frac{u}{\delta} > \frac{288}{\gamma^2\alpha\sqrt{\lambda}\tilde{\lambda}_x}$, we can get the following sufficient condition for Eq.(29) and Eq.(31)

$$T_{i,t} > \frac{288d}{\gamma^2\alpha\sqrt{\lambda}\tilde{\lambda}_x}\log(\frac{u}{\delta})\,. \tag{32}$$

Together with Eq.(24), Eq.(25), and the condition for Eq.(18) we can get the following sufficient condition for Eq.(20) to hold

$$T_{i,t} > \max\{\frac{288d}{\gamma^2\alpha\sqrt{\lambda}\tilde{\lambda}_x}\log(\frac{u}{\delta}), \frac{16}{\tilde{\lambda}_x^2}\log(\frac{8d}{\tilde{\lambda}_x^2\delta}), \frac{72\sqrt{\lambda}}{\alpha\gamma^2\tilde{\lambda}_x}, \frac{72\alpha C^2}{\gamma^2\sqrt{\lambda}\tilde{\lambda}_x}\}\,. \tag{33}$$

Then, with Assumption 2 on the uniform arrival of users, following Lemma 8 in [19], and by union bound, we can get that with probability at least $1 - \delta$, for all

$$t \geq T_0 \triangleq 16u\log(\frac{u}{\delta}) + 4u\max\{\frac{288d}{\gamma^2\alpha\sqrt{\lambda}\tilde{\lambda}_x}\log(\frac{u}{\delta}), \frac{16}{\tilde{\lambda}_x^2}\log(\frac{8d}{\tilde{\lambda}_x^2\delta}), \frac{72\sqrt{\lambda}}{\alpha\gamma^2\tilde{\lambda}_x}, \frac{72\alpha C^2}{\gamma^2\sqrt{\lambda}\tilde{\lambda}_x}\}\,, \tag{34}$$

Eq.(32) holds for all $i \in \mathcal{U}$, and therefore Eq.(20) holds for all $i \in \mathcal{U}$. With this, we can show that RCLUB-WCU will cluster all the users correctly after $T_0$. First, if RCLUB-WCU deletes the edge $(i,l)$, then user $i$ and user $j$ belong to different ground-truth clusters, i.e., $\|\boldsymbol{\theta}_i - \boldsymbol{\theta}_l\|_2 > 0$. This is because by the deletion rule of the algorithm, the concentration bound, and triangle inequality, $\|\boldsymbol{\theta}_i - \boldsymbol{\theta}_l\|_2 = \left\|\boldsymbol{\theta}^{j(i)} - \boldsymbol{\theta}^{j(l)}\right\|_2 \geq \left\|\hat{\boldsymbol{\theta}}_{i,t} - \hat{\boldsymbol{\theta}}_{l,t}\right\|_2 - \left\|\boldsymbol{\theta}^{j(l)} - \boldsymbol{\theta}_{l,t}\right\|_2 - \left\|\boldsymbol{\theta}^{j(i)} - \boldsymbol{\theta}_{i,t}\right\|_2 > 0$. Second, we show that if $\|\boldsymbol{\theta}_i - \boldsymbol{\theta}_l\| \geq \gamma$, RCLUB-WCU will delete the edge $(i,l)$. This is because if $\|\boldsymbol{\theta}_i - \boldsymbol{\theta}_l\| \geq \gamma$, then by the triangle inequality, and $\left\|\hat{\boldsymbol{\theta}}_{i,t} - \boldsymbol{\theta}^{j(i)}\right\|_2 < \frac{\gamma}{4}$, $\left\|\hat{\boldsymbol{\theta}}_{l,t} - \boldsymbol{\theta}^{j(l)}\right\|_2 < \frac{\gamma}{4}$, $\boldsymbol{\theta}_i = \boldsymbol{\theta}^{j(i)}$, $\boldsymbol{\theta}_l = \boldsymbol{\theta}^{j(l)}$, we have $\left\|\hat{\boldsymbol{\theta}}_{i,t} - \hat{\boldsymbol{\theta}}_{l,t}\right\|_2 \geq \|\boldsymbol{\theta}_i - \boldsymbol{\theta}_l\| - \left\|\hat{\boldsymbol{\theta}}_{i,t} - \boldsymbol{\theta}^{j(i)}\right\|_2 - \left\|\hat{\boldsymbol{\theta}}_{l,t} - \boldsymbol{\theta}^{j(l)}\right\|_2 > \gamma - \frac{\gamma}{4} - \frac{\gamma}{4} = \frac{\gamma}{2} > \frac{\sqrt{\lambda} + \sqrt{2\log(\frac{u}{\delta}) + d\log(1 + \frac{T_{i,t}}{\lambda d})}}{\sqrt{\lambda + 2\tilde{\lambda}_x T_{i,t}}} + \frac{\sqrt{\lambda} + \sqrt{2\log(\frac{u}{\delta}) + d\log(1 + \frac{T_{l,t}}{\lambda d})}}{\sqrt{\lambda + 2\tilde{\lambda}_x T_{l,t}}}$, which will trigger the deletion condition Line 10 in Algo.1.

## B Proof of Lemma 2

After $T_0$, if the clustering structure is correct, i.e., $V_t = V_{j(i_t)}$, then we have

$$
\begin{aligned}
\hat{\boldsymbol{\theta}}_{V_t,t-1} - \boldsymbol{\theta}_{i_t} &= \boldsymbol{M}_{V_t,t-1}^{-1}\boldsymbol{b}_{V_t,t-1} - \boldsymbol{\theta}_{i_t} \\
&= (\lambda\boldsymbol{I} + \sum_{\substack{s\in[t-1]\\i_s\in V_t}} w_{i_s,s}\boldsymbol{x}_{a_s}\boldsymbol{x}_{a_s}^\top)^{-1}(\sum_{\substack{s\in[t-1]\\i_s\in V_t}} w_{i_s,s}\boldsymbol{x}_{a_s}r_s) - \boldsymbol{\theta}_{i_t} \\
&= (\lambda\boldsymbol{I} + \sum_{\substack{s\in[t-1]\\i_s\in V_t}} w_{i_s,s}\boldsymbol{x}_{a_s}\boldsymbol{x}_{a_s}^\top)^{-1}(\sum_{\substack{s\in[t-1]\\i_s\in V_t}} w_{i_s,s}\boldsymbol{x}_{a_s}(\boldsymbol{x}_{a_s}^\top\boldsymbol{\theta}_{i_t} + \eta_s + c_s)) - \boldsymbol{\theta}_{i_t} \quad (35) \\
&= (\lambda\boldsymbol{I} + \sum_{\substack{s\in[t-1]\\i_s\in V_t}} w_{i_s,s}\boldsymbol{x}_{a_s}\boldsymbol{x}_{a_s}^\top)^{-1}\Bigg(\sum_{\substack{s\in[t-1]\\i_s\in V_t}} (w_{i_s,s}\boldsymbol{x}_{a_s}\boldsymbol{x}_{a_s}^\top + \lambda\boldsymbol{I})\boldsymbol{\theta}_{i_t} - \lambda\boldsymbol{\theta}_{i_t} \\
&\quad + \sum_{\substack{s\in[t-1]\\i_s\in V_t}} w_{i_s,s}\boldsymbol{x}_{a_s}\eta_s + \sum_{\substack{s\in[t-1]\\i_s\in V_t}} w_{i_s,s}\boldsymbol{x}_{a_s}c_s\Bigg) - \boldsymbol{\theta}_{i_t} \\
&= -\lambda\boldsymbol{M}_{V_t,t-1}'^{-1}\boldsymbol{\theta}_{i_t} - \boldsymbol{M}_{V_t,t-1}'^{-1}\sum_{\substack{s\in[t-1]\\i_s\in V_t}} w_{i_s,s}\boldsymbol{x}_{a_s}\eta_s + \boldsymbol{M}_{V_t,t-1}'^{-1}\sum_{\substack{s\in[t-1]\\i_s\in V_t}} w_{i_s,s}\boldsymbol{x}_{a_s}c_s\,,
\end{aligned}
$$

where we denote $\boldsymbol{M}_{V_t,t-1}' = \boldsymbol{M}_{V_t,t-1} + \lambda\boldsymbol{I}$, and Eq.(35) is because $V_t = V_{j(i_t)}$ thus $\boldsymbol{\theta}_{i_s} = \boldsymbol{\theta}_{i_t}, \forall i_s \in V_t$.

Therefore, we have

$$
\begin{aligned}
\left|\boldsymbol{x}_a^\top(\hat{\boldsymbol{\theta}}_{V_t,t-1} - \boldsymbol{\theta}_{i_t})\right| &\leq \lambda\left|\boldsymbol{x}_a^\top\boldsymbol{M}_{V_t,t-1}'^{-1}\boldsymbol{\theta}_{i_t}\right| + \left|\boldsymbol{x}_a^\top\boldsymbol{M}_{V_t,t-1}'^{-1}\sum_{\substack{s\in[t-1]\\i_s\in V_t}} w_{i_s,s}\boldsymbol{x}_{a_s}\eta_s\right| + \left|\boldsymbol{x}_a^\top\boldsymbol{M}_{V_t,t-1}'^{-1}\sum_{\substack{s\in[t-1]\\i_s\in V_t}} w_{i_s,s}\boldsymbol{x}_{a_s}c_s\right| \\
&\leq \|\boldsymbol{x}_a\|_{\boldsymbol{M}_{V_t,t-1}'^{-1}}\Bigg(\sqrt{\lambda} + \left\|\sum_{\substack{s\in[t-1]\\i_s\in V_t}} w_{i_s,s}\boldsymbol{x}_{a_s}\eta_s\right\|_{\boldsymbol{M}_{V_t,t-1}'^{-1}} + \left\|\sum_{\substack{s\in[t-1]\\i_s\in V_t}} w_{i_s,s}\boldsymbol{x}_{a_s}c_s\right\|_{\boldsymbol{M}_{V_t,t-1}'^{-1}}\Bigg)\,,
\end{aligned}
$$
(36)

where Eq.(36) is by Cauchy–Schwarz inequality, matrix operator inequality, and $\left|\boldsymbol{x}_a^\top\boldsymbol{M}_{V_t,t-1}'^{-1}\boldsymbol{\theta}_{i_t}\right| \leq \lambda\left\|\boldsymbol{M}_{V_t,t-1}'^{-\frac{1}{2}}\right\|_2\|\boldsymbol{\theta}_{i_t}\|_2 = \lambda\frac{1}{\sqrt{\lambda_{\min}(\boldsymbol{M}_{V_t,t-1})}}\|\boldsymbol{\theta}_{i_t}\|_2 \leq \sqrt{\lambda}$ since $\lambda_{\min}(\boldsymbol{M}_{V_t,t-1}) \geq \lambda$ and $\|\boldsymbol{\theta}_{i_t}\|_2 \leq 1$.

Let $\tilde{\boldsymbol{x}}_s \triangleq \sqrt{w_{i_s,s}}\boldsymbol{x}_{a_s}$, $\tilde{\eta}_s \triangleq \sqrt{w_{i_s,s}}\eta_s$, then we have: $\|\tilde{\boldsymbol{x}}_s\|_2 \leq \left\|\sqrt{w_{i_s,s}}\right\|_2\|\boldsymbol{x}_{a_s}\|_2 \leq 1$, $\tilde{\eta}_s$ is still 1-sub-gaussian (since $\eta_s$ is 1-sub-gaussian and $\sqrt{w_{i_s,s}} \leq 1$), $\boldsymbol{M}_{i,t}' = \lambda\boldsymbol{I} + \sum_{\substack{s\in[t]\\i_s=i}} \tilde{\boldsymbol{x}}_s\tilde{\boldsymbol{x}}_s^\top$,

and $\left\|\sum_{\substack{s\in[t-1]\\i_s\in V_t}} w_{i_s,s}\boldsymbol{x}_{a_s}\eta_s\right\|_{\boldsymbol{M}'^{-1}_{V_t,t-1}}$ becomes $\left\|\sum_{\substack{s\in[t]\\i_s=i}} \tilde{\boldsymbol{x}}_s\tilde{\eta}_s\right\|_{\boldsymbol{M}'^{-1}_{V_t,t-1}}$ . Then, following Theorem 1 in [1], with probability at least $1-\delta$ for some $\delta\in(0,1)$, we have:

$$
\left\|\sum_{\substack{s\in[t-1]\\i_s\in V_t}} w_{i_s,s}\boldsymbol{x}_{a_s}\eta_s\right\|_{\boldsymbol{M}'^{-1}_{V_t,t-1}} = \left\|\sum_{\substack{s\in[t]\\i_s=i}} \tilde{\boldsymbol{x}}_s\tilde{\eta}_s\right\|_{\boldsymbol{M}'^{-1}_{V_t,t-1}}
$$

$$
\leq \sqrt{2\log(\frac{u}{\delta}) + \log(\frac{\det(\boldsymbol{M}'_{V_t,t-1})}{\det(\lambda\boldsymbol{I})})}
$$

$$
\leq \sqrt{2\log(\frac{u}{\delta}) + d\log(1+\frac{T}{\lambda d})}, \tag{37}
$$

And for $\left\|\sum_{\substack{s\in[t-1]\\i_s\in V_t}} w_{i_s,s}\boldsymbol{x}_{a_s}c_s\right\|_{\boldsymbol{M}'^{-1}_{V_t,t-1}}$ , we have

$$
\left\|\sum_{\substack{s\in[t-1]\\i_s\in V_t}} w_{i_s,s}\boldsymbol{x}_{a_s}c_s\right\|_{\boldsymbol{M}'^{-1}_{V_t,t-1}} \leq \sum_{\substack{s\in[t-1]\\i_s\in V_t}} w_{i_s,s}\,|c_s|\,\|\boldsymbol{x}_{a_s}\|_{\boldsymbol{M}'^{-1}_{V_t,t-1}} \leq \alpha C, \tag{38}
$$

where we use $\sum_{t=1}^T |c_t| \leq C$, $w_{i_s,s} \leq \frac{\alpha}{\|\boldsymbol{x}_{a_s}\|_{\boldsymbol{M}'^{-1}_{i_s,t-1}}} \leq \frac{\alpha}{\|\boldsymbol{x}_{a_s}\|_{\boldsymbol{M}'^{-1}_{V_t,t-1}}}$ .

Plugging Eq.(38) and Eq.(37) into Eq.(36), together with Lemma 1, we can complete the proof of Lemma 2.

## C  Proof of Theorem 3

After $T_0$, we define event

$$
\mathcal{E} = \{\text{the algorithm clusters all the users correctly for all } t \geq T_0\}. \tag{39}
$$

Then, with Lemma 1 and picking $\delta = \frac{1}{T}$, we have

$$
R(T) = \mathbb{P}(\mathcal{E})\mathbb{I}\{\mathcal{E}\}R(T) + \mathbb{P}(\overline{\mathcal{E}})\mathbb{I}\{\overline{\mathcal{E}}\}R(T)
$$

$$
\leq \mathbb{I}\{\mathcal{E}\}R(T) + 4 \times \frac{1}{T} \times T \tag{40}
$$

$$
= \mathbb{I}\{\mathcal{E}\}R(T) + 4.
$$

Then it remains to bound $\mathbb{I}\{\mathcal{E}\}R(T)$. For the first $T_0$ rounds, we can upper bound the regret in the first $T_0$ rounds by $T_0$. After $T_0$, under event $\mathcal{E}$ and by Lemma 2, we have that with probability at least $1-\delta$, for any $\boldsymbol{x}_a$:

$$
\left|\boldsymbol{x}_a^{\mathrm{T}}(\hat{\boldsymbol{\theta}}_{V_t,t-1} - \boldsymbol{\theta}_{i_t})\right| \leq \beta\,\|\boldsymbol{x}_a\|_{\boldsymbol{M}^{-1}_{V_t,t-1}} \triangleq C_{a,t}. \tag{41}
$$

Therefore, for the instantaneous regret $R_t$ at round $t$, with $\mathcal{E}$, with probability at least $1-\delta$, at $\forall t \geq T_0$:

$$
R_t = \boldsymbol{x}_{a_t^*}^{\top}\boldsymbol{\theta}_{i_t} - \boldsymbol{x}_{a_t}^{\top}\boldsymbol{\theta}_{i_t}
$$

$$
= \boldsymbol{x}_{a_t^*}^{\top}(\boldsymbol{\theta}_{i_t} - \hat{\boldsymbol{\theta}}_{V_t,t-1}) + (\boldsymbol{x}_{a_t^*}^{\top}\hat{\boldsymbol{\theta}}_{V_t,t-1} + C_{a_t^*,t}) - (\boldsymbol{x}_{a_t}^{\top}\hat{\boldsymbol{\theta}}_{V_t,t-1} + C_{a_t,t}) \tag{42}
$$

$$
+ \boldsymbol{x}_{a_t}^{\top}(\hat{\boldsymbol{\theta}}_{\overline{V}_t,t-1} - \boldsymbol{\theta}_{i_t}) + C_{a_t,t} - C_{a_t^*,t}
$$

$$
\leq 2C_{a_t,t},
$$

where the last inequality holds by the UCB arm selection strategy in Eq.(3) and Eq.(41).

Therefore, for $\mathbb{I}\{\mathcal{E}\}R(T)$:

$$\mathbb{I}\{\mathcal{E}\}R(T) \leq R(T_0) + \mathbb{E}[\mathbb{I}\{\mathcal{E}\} \sum_{t=T_0+1}^{T} R_t]$$

$$\leq T_0 + 2\mathbb{E}[\mathbb{I}\{\mathcal{E}\} \sum_{t=T_0+1}^{T} C_{a_t,t}]. \tag{43}$$

Then it remains to bound $\mathbb{E}[\mathbb{I}\{\mathcal{E}\} \sum_{t=T_0+1}^{T} C_{a_t,t}]$. For $\sum_{t=T_0+1}^{T} C_{a_t,t}$, we can distinguish it into two cases:

$$\sum_{t=T_0+1}^{T} C_{a_t,t} \leq \beta \sum_{t=1}^{T} \|\boldsymbol{x}_{\boldsymbol{a_t}}\|_{\boldsymbol{M}_{V_t,t-1}^{-1}}$$

$$= \beta \sum_{t\in[T]:w_{i_t,t}=1} \|\boldsymbol{x}_{\boldsymbol{a_t}}\|_{\boldsymbol{M}_{V_t,t-1}^{-1}} + \beta \sum_{t\in[T]:w_{i_t,t}<1} \|\boldsymbol{x}_{\boldsymbol{a_t}}\|_{\boldsymbol{M}_{V_t,t-1}^{-1}}. \tag{44}$$

Then, we prove the following technical lemma.

**Lemma 7.**

$$\sum_{t=T_0+1}^{T} \min\{\mathbb{I}\{i_t \in V_j\} \|\boldsymbol{x}_{a_t}\|_{\boldsymbol{M}_{V_j,t-1}^{-1}}^2, 1\} \leq 2d \log(1 + \frac{T}{\lambda d}), \forall j \in [m]. \tag{45}$$

*Proof.*

$$det(\boldsymbol{M}_{V_j,T}) = det\left(\boldsymbol{M}_{V_j,T-1} + \mathbb{I}\{i_T \in V_j\}\boldsymbol{x}_{a_T}\boldsymbol{x}_{a_T}^\top\right)$$

$$= det(\boldsymbol{M}_{V_j,T-1})det\left(\boldsymbol{I} + \mathbb{I}\{i_T \in V_j\}\boldsymbol{M}_{V_j,T-1}^{-\frac{1}{2}}\boldsymbol{x}_{a_T}\boldsymbol{x}_{a_T}^\top\boldsymbol{M}_{V_j,T-1}^{-\frac{1}{2}}\right)$$

$$= det(\boldsymbol{M}_{V_j,T-1})\left(1 + \mathbb{I}\{i_T \in V_j\} \|\boldsymbol{x}_{a_T}\|_{\boldsymbol{M}_{V_j,T-1}^{-1}}^2\right)$$

$$= det(\boldsymbol{M}_{V_j,T_0}) \prod_{t=T_0+1}^{T} \left(1 + \mathbb{I}\{i_t \in V_j\} \|\boldsymbol{x}_{a_t}\|_{\boldsymbol{M}_{V_j,t-1}^{-1}}^2\right)$$

$$\geq det(\lambda\boldsymbol{I}) \prod_{t=T_0+1}^{T} \left(1 + \mathbb{I}\{i_t \in V_j\} \|\boldsymbol{x}_{a_t}\|_{\boldsymbol{M}_{V_j,t-1}^{-1}}^2\right). \tag{46}$$

$\forall x \in [0,1]$, we have $x \leq 2\log(1+x)$. Therefore

$$\sum_{t=T_0+1}^{T} \min\{\mathbb{I}\{i_t \in V_j\} \|\boldsymbol{x}_{a_t}\|_{\boldsymbol{M}_{V_j,t-1}^{-1}}^2, 1\} \leq 2 \sum_{t=T_0+1}^{T} \log\left(1 + \mathbb{I}\{i_t \in V_j\} \|\boldsymbol{x}_{a_t}\|_{\boldsymbol{M}_{V_j,t-1}^{-1}}^2\right)$$

$$= 2\log\left(\prod_{t=T_0+1}^{T} \left(1 + \mathbb{I}\{i_t \in V_j\} \|\boldsymbol{x}_{a_t}\|_{\boldsymbol{M}_{V_j,t-1}^{-1}}^2\right)\right)$$

$$\leq 2[\log(det(\boldsymbol{M}_{V_j,T})) - \log(det(\lambda\boldsymbol{I}))]$$

$$\leq 2\log\left(\frac{trace(\lambda\boldsymbol{I} + \sum_{t=1}^{T} \mathbb{I}\{i_t \in V_j\}\boldsymbol{x}_{a_t}\boldsymbol{x}_{a_t}^\top)}{\lambda d}\right)^d$$

$$\leq 2d\log(1 + \frac{T}{\lambda d}). \tag{47}$$

$\square$

Denote the rounds with $w_{i_t,t} = 1$ as $\{\tilde{t}_1, \ldots, \tilde{t}_{l_1}\}$, and gram matrix $\tilde{G}_{V_{\tilde{t}_\tau}, \tilde{t}_\tau - 1} \triangleq \lambda I + \sum_{\substack{s \in [\tau] \\ i_s \in V_{\tilde{t}_\tau}}} x_{a_{\tilde{t}_s}} x_{a_{\tilde{t}_s}}^\top$; denote the rounds with $w_{i_t,t} < 1$ as $\{t'_1, \ldots, t'_{l_2}\}$, gram matrix $G'_{V_{t'_\tau}, t'_\tau - 1} \triangleq \lambda I + \sum_{\substack{s \in [\tau] \\ i_s \in V_{t'_\tau}}} w_{i_{t'_s}, t'_s} x_{a_{t'_s}} x_{a_{t'_s}}^\top$.

Then we have

$$
\sum_{t \in [T]: w_{i_t,t}=1} \|x_{a_t}\|_{M_{V_t,t-1}^{-1}} = \sum_{j=1}^m \sum_{\tau=1}^{l_1} \mathbb{I}\{i_{\tilde{t}_\tau} \in V_j\} \left\|x_{a_{\tilde{t}_\tau}}\right\|_{M_{V_{\tilde{t}_\tau}, \tilde{t}_\tau - 1}^{-1}} \leq \sum_{j=1}^m \sum_{\tau=1}^{l_1} \mathbb{I}\{i_{\tilde{t}_\tau} \in V_j\} \left\|x_{a_{\tilde{t}_\tau}}\right\|_{\tilde{G}_{V_{\tilde{t}_\tau}, \tilde{t}_\tau - 1}^{-1}}
$$
(48)

$$
\leq \sum_{j=1}^m \sqrt{\sum_{\tau=1}^{l_1} \mathbb{I}\{i_{\tilde{t}_\tau} \in V_j\} \sum_{\tau=1}^{l_1} \min\{1, \mathbb{I}\{i_{\tilde{t}_\tau} \in V_j\} \left\|x_{a_{\tilde{t}_\tau}}\right\|_{\tilde{G}_{V_{\tilde{t}_\tau}, \tilde{t}_\tau - 1}^{-1}}^2\}}
$$
(49)

$$
\leq \sum_{j=1}^m \sqrt{T_{V_j,T} \times 2d \log(1 + \frac{T}{\lambda d})}
$$
(50)

$$
\leq \sqrt{2m \sum_{j=1}^m T_{V_j,T} d \log(1 + \frac{T}{\lambda d})} = \sqrt{2mdT \log(1 + \frac{T}{\lambda d})},
$$
(51)

where Eq.(48) is because $\tilde{G}_{V_{\tilde{t}_\tau}, \tilde{t}_\tau - 1}^{-1} \succeq M_{V_{\tilde{t}_\tau}, \tilde{t}_\tau - 1}^{-1}$ in Eq.(49) we use Cauchy–Schwarz inequality, in Eq.(50) we use Lemma 7 and $\sum_{\tau=1}^{l_1} \mathbb{I}\{i_{\tilde{t}_\tau} \in V_j\} \leq T_{V_j,T}$, in Eq.(51) we use Cauchy–Schwarz inequality and $\sum_{j=1}^m T_{V_j,T} = T$.

For the second part in Eq.(44), Let $x'_{a_{t'_\tau}} \triangleq \sqrt{w_{i_{t'_\tau}, t'_\tau}} x_{a_{t'_\tau}}$, then

$$
\sum_{t: w_{i_t,t} < 1} \|x_{a_t}\|_{M_{V_t,t-1}^{-1}} = \sum_{t: w_{i_t,t} < 1} \frac{\|x_{a_t}\|_{M_{V_t,t-1}^{-1}}^2}{\|x_{a_t}\|_{M_{V_t,t-1}^{-1}}} = \sum_{t: w_{i_t,t} < 1} \frac{w_{i_t,t} \|x_{a_t}\|_{M_{V_t,t-1}^{-1}}^2}{\alpha}
$$
(52)

$$
= \sum_{j=1}^m \sum_{\tau=1}^{l_2} \mathbb{I}\{i_{t'_\tau} \in V_j\} \frac{w_{i_{t'_\tau}, t'_\tau}}{\alpha} \left\|x_{a_{t'_\tau}}\right\|_{M_{V_{t'_\tau}, t'_\tau - 1}^{-1}}^2
$$

$$
\leq \sum_{j=1}^m \frac{\sum_{\tau=1}^{l_2} \min\{1, \mathbb{I}\{i_{t'_\tau} \in V_j\} \left\|x'_{a_{t'_\tau}}\right\|_{G'^{-1}_{V_{t'_\tau}, t'_\tau - 1}}^2\}}{\alpha}
$$
(53)

$$
\leq \sum_{j=1}^m \frac{2d \log(1 + \frac{T}{\lambda d})}{\alpha} = \frac{2md \log(1 + \frac{T}{\lambda d})}{\alpha}
$$
(54)

where in Eq.(52) we use the definition of the weights, in Eq.(53) we use $G'^{-1}_{V_{t'_\tau}, t'_\tau - 1} \succeq M_{V_{t'_\tau}, t'_\tau - 1}^{-1}$, and Eq.(54) uses Lemma 7.

Then, with Eq.(54), Eq.(51), Eq.(44), Eq.(40), Eq.(43), $\delta = \frac{1}{T}$, and $\beta = \sqrt{\lambda} + \sqrt{2\log(T) + d\log(1 + \frac{T}{\lambda d})} + \alpha C$, we can get

$$
\begin{aligned}
R(T) \leq {} & 4 + T_0 + \left(2\sqrt{\lambda} + \sqrt{2\log(T) + d\log(1 + \frac{T}{\lambda d})} + \alpha C\right) \times \left(\sqrt{2mdT\log(1 + \frac{T}{\lambda d})}\right. \\
& \left. + \frac{2md\log(1 + \frac{T}{\lambda d})}{\alpha}\right) \\
= {} & 4 + 16u\log(uT) + 4u\max\{\frac{288d}{\gamma^2\alpha\sqrt{\lambda}\tilde{\lambda}_x}\log(uT), \frac{16}{\tilde{\lambda}_x^2}\log(\frac{8dT}{\tilde{\lambda}_x^2}), \frac{72\sqrt{\lambda}}{\alpha\gamma^2\tilde{\lambda}_x}, \frac{72\alpha C^2}{\gamma^2\sqrt{\lambda}\tilde{\lambda}_x}\} \\
& + \left(2\sqrt{\lambda} + \sqrt{2\log(T) + d\log(1 + \frac{T}{\lambda d})} + \alpha C\right) \times \left(\sqrt{2mdT\log(1 + \frac{T}{\lambda d})}\right. \\
& \left. + \frac{2md\log(1 + \frac{T}{\lambda d})}{\alpha}\right).
\end{aligned}
$$

Picking $\alpha = \frac{\sqrt{\lambda} + \sqrt{d}}{C}$, we can get

$$
R(T) \leq O\left(\left(\frac{C\sqrt{d}}{\gamma^2\tilde{\lambda}_x} + \frac{1}{\tilde{\lambda}_x^2}\right)u\log(T)\right) + O\left(d\sqrt{mT}\log(T)\right) + O\left(mCd\log^{1.5}(T)\right). \tag{55}
$$

Thus we complete the proof of Theorem 3.

## D  Proof and Discussions of Theorem 4

Table 1 of the work [12] gives a lower bound for linear bandits with adversarial corruption for a single user. The lower bound of $R(T)$ is given by: $R(T) \geq \Omega(d\sqrt{T} + dC)$. Therefore, suppose our problem with multiple users and $m$ underlying clusters where the arrival times are $T_i$ for each cluster, then for any algorithms, even if they know the underlying clustering structure and keep $m$ independent linear bandit algorithms to leverage the common information of clusters, the best they can get is $R(T) \geq dC + \sum_{i \in [m]} d\sqrt{T_i}$. For a special case where $T_i = \frac{T}{m}, \forall i \in [m]$, we can get $R(T) \geq dC + \sum_{i \in [m]} d\sqrt{\frac{T}{m}} = d\sqrt{mT} + dC$, which gives a lower bound of $\Omega(d\sqrt{mT} + dC)$ for the LOCUD problem.

Recall that the regret upper bound of RCLUB-WCU shown in Theorem 3 is of $O\left(\left(\frac{C\sqrt{d}}{\gamma^2\tilde{\lambda}_x} + \frac{1}{\tilde{\lambda}_x^2}\right)u\log(T)\right) + O\left(d\sqrt{mT}\log(T)\right) + O\left(mCd\log^{1.5}(T)\right)$, asymptotically matching this lower bound with respect to $T$ up to logarithmic factors and with respect to $C$ up to $O(\sqrt{m})$ factors, showing the tightness of our theoretical results (where $m$ are typically very small for real applications).

We conjecture that the gap for the $m$ factor in the $mC$ term of the lower bound is due to the strong assumption that cluster structures are known to prove our lower bound, and whether there exists a tighter lower bound will be left for future work.

## E  Proof of Theorem 5

We prove the theorem using the proof by contrapositive. Specifically, in Theorem 5, we need to prove that for any $t \geq T_0$, if the detection condition in Line 7 of Algo.2 for user $i$, then with probability at least $1 - 5\delta$, user $i$ is indeed a corrupted user. By the proof by contrapositive, we can prove Theorem 5 by showing that: for any $t \geq T_0$, if user $i$ is a normal user, then with probability at least $1 - 5\delta$, the detection condition in Line 7 of Algo.2 will not be satisfied for user $i$.

If the clustering structure is correct at $t$, then for any normal user $i$

$$
\tilde{\boldsymbol{\theta}}_{i,t} - \hat{\boldsymbol{\theta}}_{V_{i,t},t} = \tilde{\boldsymbol{\theta}}_{i,t} - \boldsymbol{\theta}_i + \boldsymbol{\theta}_i - \hat{\boldsymbol{\theta}}_{V_{i,t},t}, \tag{56}
$$

where $\tilde{\boldsymbol{\theta}}_{i,t}$ is the non-robust estimation of the ground-truth $\theta_i$, and $\hat{\boldsymbol{\theta}}_{V_{i,t},t-1}$ is the robust estimation of the inferred cluster $V_{i,t}$ for user $i$ at round $t$. Since the clustering structure is correct at $t$, $\hat{\boldsymbol{\theta}}_{V_{i,t},t-1}$ is the robust estimation of user $i$'s ground-truth cluster's preference vector $\boldsymbol{\theta}^{j(i)} = \boldsymbol{\theta}_i$ at round $t$.

We have

$$
\begin{aligned}
\tilde{\boldsymbol{\theta}}_{i,t} - \boldsymbol{\theta}_i &= (\lambda \boldsymbol{I} + \tilde{\boldsymbol{M}}_{i,t})^{-1} \tilde{\boldsymbol{b}}_{i,t} - \boldsymbol{\theta}_i \\
&= (\lambda \boldsymbol{I} + \sum_{\substack{s \in [t] \\ i_s = i}} \boldsymbol{x}_{a_s} \boldsymbol{x}_{a_s}^\top)^{-1} (\sum_{\substack{s \in [t] \\ i_s = i}} \boldsymbol{x}_{a_s} r_s) - \boldsymbol{\theta}_i \\
&= (\lambda \boldsymbol{I} + \sum_{\substack{s \in [t] \\ i_s = i}} \boldsymbol{x}_{a_s} \boldsymbol{x}_{a_s}^\top)^{-1} (\sum_{\substack{s \in [t] \\ i_s = i}} \boldsymbol{x}_{a_s} (\boldsymbol{x}_{a_s}^\top \boldsymbol{\theta}_i + \eta_s)) - \boldsymbol{\theta}_i \quad (57) \\
&= (\lambda \boldsymbol{I} + \sum_{\substack{s \in [t] \\ i_s = i}} \boldsymbol{x}_{a_s} \boldsymbol{x}_{a_s}^\top)^{-1} \Big( (\lambda \boldsymbol{I} + \sum_{\substack{s \in [t] \\ i_s = i}} \boldsymbol{x}_{a_s} \boldsymbol{x}_{a_s}^\top) \boldsymbol{\theta}_i - \lambda \boldsymbol{\theta}_i + \sum_{\substack{s \in [t] \\ i_s = i}} \boldsymbol{x}_{a_s} \eta_s \Big) - \boldsymbol{\theta}_i \\
&= -\lambda \tilde{\boldsymbol{M}}_{i,t}^{'-1} \boldsymbol{\theta}_i + \tilde{\boldsymbol{M}}_{i,t}^{'-1} \sum_{\substack{s \in [t] \\ i_s = i}} \boldsymbol{x}_{a_s} \eta_s \,,
\end{aligned}
$$

where we denote $\tilde{\boldsymbol{M}}_{i,t}^{'} \triangleq \lambda \boldsymbol{I} + \sum_{\substack{s \in [t] \\ i_s = i}} \boldsymbol{x}_{a_s} \boldsymbol{x}_{a_s}^\top$, and Eq.(57) is because since user $i$ is normal, we have $c_s = 0, \forall s : i_s = i$.

Then, we have

$$
\begin{aligned}
\left\| \tilde{\boldsymbol{\theta}}_{i,t} - \boldsymbol{\theta}_i \right\|_2 &\le \left\| \lambda \tilde{\boldsymbol{M}}_{i,t}^{'-1} \boldsymbol{\theta}_i \right\|_2 + \left\| \tilde{\boldsymbol{M}}_{i,t}^{'-1} \sum_{\substack{s \in [t] \\ i_s = i}} \boldsymbol{x}_{a_s} \eta_s \right\|_2 \\
&\le \lambda \left\| \tilde{\boldsymbol{M}}_{i,t}^{'-\frac{1}{2}} \right\|_2^2 \|\boldsymbol{\theta}_i\|_2 + \left\| \tilde{\boldsymbol{M}}_{i,t}^{'-\frac{1}{2}} \sum_{\substack{s \in [t] \\ i_s = i}} \boldsymbol{x}_{a_s} \eta_s \right\|_2 \left\| \tilde{\boldsymbol{M}}_{i,t}^{'-\frac{1}{2}} \right\|_2 \quad (58) \\
&\le \frac{\sqrt{\lambda} + \left\| \sum_{\substack{s \in [t] \\ i_s = i}} \boldsymbol{x}_{a_s} \eta_s \right\|_{\tilde{\boldsymbol{M}}_{i,t}^{'-1}}}{\sqrt{\lambda_{\min}(\tilde{\boldsymbol{M}}_{i,t}^{'})}} \,, \,, \quad (59)
\end{aligned}
$$

where Eq.(58) follows by the Cauchy–Schwarz inequality and the inequality for the operator norm of matrices, and Eq.(59) follows by the Courant-Fischer theorem and the fact that $\lambda_{\min}(\tilde{\boldsymbol{M}}_{i,t}^{'}) \ge \lambda$.

Following Theorem 1 in [1], for a fixed normal user $i$, with probability at least $1 - \delta$ for some $\delta \in (0, 1)$ we have:

$$
\begin{aligned}
\left\| \sum_{\substack{s \in [t] \\ i_s = i}} \boldsymbol{x}_{a_s} \eta_s \right\|_{\tilde{\boldsymbol{M}}_{i,t}^{'-1}} &\le \sqrt{2 \log(\frac{1}{\delta}) + \log(\frac{\det(\tilde{\boldsymbol{M}}_{i,t}^{'})}{\det(\lambda \boldsymbol{I})})} \\
&\le \sqrt{2 \log(\frac{1}{\delta}) + d \log(1 + \frac{T_{i,t}}{\lambda d})} \,, \quad (60)
\end{aligned}
$$

where Eq.(60) is because $\det(\tilde{\boldsymbol{M}}_{i,t}^{'}) \le \left( \frac{\text{trace}(\lambda \boldsymbol{I} + \sum_{\substack{s \in [t] \\ i_s = i}} \boldsymbol{x}_{a_s} \boldsymbol{x}_{a_s}^\top)}{d} \right)^d \le \left( \frac{\lambda d + T_{i,t}}{d} \right)^d$, and $\det(\lambda \boldsymbol{I}) = \lambda^d$.

Plugging this into Eq.(59), we can get

$$
\left\| \tilde{\boldsymbol{\theta}}_{i,t} - \boldsymbol{\theta}_i \right\|_2 \le \frac{\sqrt{\lambda} + \sqrt{2 \log(\frac{1}{\delta}) + d \log(1 + \frac{T_{i,t}}{\lambda d})}}{\sqrt{\lambda_{\min}(\tilde{\boldsymbol{M}}_{i,t}^{'})}} \,. \quad (61)
$$

Then we need to bound $\left\|\boldsymbol{\theta}_i - \hat{\boldsymbol{\theta}}_{V_{i,t},t}\right\|_2$. With the correct clustering, $V_{i,t} = V_{j(i)}$, we have

$$
\hat{\boldsymbol{\theta}}_{V_{i,t},t} - \boldsymbol{\theta}_i = M_{V_{i,t},t}^{-1} b_{V_{j,t},t}
$$

$$
= (\lambda I + \sum_{\substack{s \in [t] \\ i_s \in V_{j(i)}}} w_{i_s,s} \boldsymbol{x}_{a_s} \boldsymbol{x}_{a_s}^{\top})^{-1} (\sum_{\substack{s \in [t] \\ i_s \in V_{j(i)}}} w_{i_s,s} \boldsymbol{x}_{a_s} r_s) - \theta_i
$$

$$
= (\lambda I + \sum_{\substack{s \in [t] \\ i_s \in V_{j(i)}}} w_{i_s,s} \boldsymbol{x}_{a_s} \boldsymbol{x}_{a_s}^{\top})^{-1} (\sum_{\substack{s \in [t] \\ i_s \in V_{j(i)}}} w_{i_s,s} \boldsymbol{x}_{a_s} (\boldsymbol{x}_{a_s}^{\top} \boldsymbol{\theta}_i + \eta_s + c_s))) - \theta_i \quad (62)
$$

$$
= (\lambda I + \sum_{\substack{s \in [t] \\ i_s \in V_{j(i)}}} w_{i_s,s} \boldsymbol{x}_{a_s} \boldsymbol{x}_{a_s}^{\top})^{-1} \big((\lambda I + \sum_{\substack{s \in [t] \\ i_s \in V_{j(i)}}} w_{i_s,s} \boldsymbol{x}_{a_s} \boldsymbol{x}_{a_s}^{\top}) \boldsymbol{\theta}_i - \lambda \boldsymbol{\theta}_i
$$

$$
+ \sum_{\substack{s \in [t] \\ i_s \in V_{j(i)}}} w_{i_s,s} \boldsymbol{x}_{a_s} \eta_s + \sum_{\substack{s \in [t] \\ i_s \in V_{j(i)}}} w_{i_s,s} \boldsymbol{x}_{a_s} c_s)) - \theta_i
$$

$$
= -\lambda M_{V_{i,t},t}^{-1} \boldsymbol{\theta}_i + M_{V_{i,t},t}^{-1} \sum_{\substack{s \in [t] \\ i_s \in V_{j(i)}}} w_{i_s,s} \boldsymbol{x}_{a_s} \eta_s + M_{V_{i,t},t}^{-1} \sum_{\substack{s \in [t] \\ i_s \in V_{j(i)}}} w_{i_s,s} \boldsymbol{x}_{a_s} c_s . \quad (63)
$$

Therefore, we have

$$
\left\|\boldsymbol{\theta}_i - \hat{\boldsymbol{\theta}}_{V_{i,t},t}\right\|_2 \le \lambda \left\|M_{V_{i,t},t}^{-1} \boldsymbol{\theta}_i\right\|_2 + \left\|M_{V_{i,t},t}^{-1} \sum_{\substack{s \in [t] \\ i_s \in V_{j(i)}}} w_{i_s,s} \boldsymbol{x}_{a_s} \eta_s\right\|_2 + \left\|M_{V_{i,t},t}^{-1} \sum_{\substack{s \in [t] \\ i_s \in V_{j(i)}}} w_{i_s,s} \boldsymbol{x}_{a_s} c_s\right\|_2
$$

$$
\le \lambda \left\|M_{V_{i,t},t}^{-\frac{1}{2}}\right\|_2^2 \|\boldsymbol{\theta}_i\|_2 + \left\|M_{V_{i,t},t}^{-\frac{1}{2}} \sum_{\substack{s \in [t] \\ i_s \in V_{j(i)}}} w_{i_s,s} \boldsymbol{x}_{a_s} \eta_s\right\|_2 \left\|M_{V_{i,t},t}^{-\frac{1}{2}}\right\|_2
$$

$$
+ \left\|M_{V_{i,t},t}^{-\frac{1}{2}} \sum_{\substack{s \in [t] \\ i_s \in V_{j(i)}}} w_{i_s,s} \boldsymbol{x}_{a_s} \eta_s\right\|_2 \left\|M_{V_{i,t},t}^{-\frac{1}{2}}\right\|_2 \quad (64)
$$

$$
\le \frac{\sqrt{\lambda} + \left\|\sum_{\substack{s \in [t] \\ i_s \in V_{j(i)}}} w_{i_s,s} \boldsymbol{x}_{a_s} \eta_s\right\|_{M_{V_{i,t},t}^{-1}} + \left\|\sum_{\substack{s \in [t] \\ i_s \in V_{j(i)}}} w_{i_s,s} \boldsymbol{x}_{a_s} c_s\right\|_{M_{V_{i,t},t}^{-1}}}{\sqrt{\lambda_{\min}(M_{V_{i,t},t})}} \quad (65)
$$

Let $\tilde{\boldsymbol{x}}_s \triangleq \sqrt{w_{i_s,s}} \boldsymbol{x}_{a_s}$, $\tilde{\eta}_s \triangleq \sqrt{w_{i_s,s}} \eta_s$, then we have: $\|\tilde{\boldsymbol{x}}_s\|_2 \le \left\|\sqrt{w_{i_s,s}}\right\|_2 \|\boldsymbol{x}_{a_s}\|_2 \le 1$, $\tilde{\eta}_s$ is still 1-sub-gaussian (since $\eta_s$ is 1-sub-gaussian and $\sqrt{w_{i_s,s}} \le 1$), $M_{V_{i,t},t} = \lambda I + \sum_{\substack{s \in [t] \\ i_s \in V_{j(i)}}} \tilde{\boldsymbol{x}}_s \tilde{\boldsymbol{x}}_s^{\top}$, and $\left\|\sum_{\substack{s \in [t] \\ i_s \in V_{j(i)}}} w_{i_s,s} \boldsymbol{x}_{a_s} \eta_s\right\|_{M_{V_{i,t},t}^{-1}}$ becomes $\left\|\sum_{\substack{s \in [t] \\ i_s \in V_{j(i)}}} \tilde{\boldsymbol{x}}_s \tilde{\eta}_s\right\|_{M_{V_{i,t},t}^{-1}}$. Then, following Theorem 1 in [1], with probability at least $1 - \delta$ for some $\delta \in (0, 1)$, for a fixed normal user $i$, we have

$$
\left\|\sum_{\substack{s \in [t] \\ i_s \in V_{j(i)}}} w_{i_s,s} \boldsymbol{x}_{a_s} \eta_s\right\|_{M_{V_{i,t},t}^{-1}} \le \sqrt{2 \log(\frac{1}{\delta}) + \log(\frac{\det(M_{V_{i,t},t})}{\det(\lambda I)})}
$$

$$
\le \sqrt{2 \log(\frac{1}{\delta}) + d \log(1 + \frac{T_{V_{i,t},t}}{\lambda d})}, \quad (66)
$$

where Eq.(60) is because $\det(M_{V_{i,t},t}) \le \left(\frac{\text{trace}(\lambda I + \sum_{\substack{s \in [t] \\ i_s \in V_{j(i)}}} \boldsymbol{x}_{a_s} \boldsymbol{x}_{a_s}^{\top})}{d}\right)^d \le \left(\frac{\lambda d + T_{V_{i,t},t}}{d}\right)^d$, and $\det(\lambda I) = \lambda^d$.

For $\left\|\sum_{\substack{s\in[t]\\ i_s\in V_{j(i)}}} w_{i_s,s}\boldsymbol{x}_{a_s}c_s\right\|_{\boldsymbol{M}_{V_{i,t},t}^{-1}}$, we have

$$\left\|\sum_{\substack{s\in[t]\\ i_s\in V_{j(i)}}} w_{i_s,s}\boldsymbol{x}_{a_s}c_s\right\|_{\boldsymbol{M}_{V_{i,t},t}^{-1}} \leq \sum_{\substack{s\in[t]\\ i_s\in V_{j(i)}}} |c_s|\, w_{i_s,s}\|\boldsymbol{x}_{a_s}\|_{\boldsymbol{M}_{V_{i,t},t}^{-1}}$$

$$\leq \alpha C\,, \tag{67}$$

where Eq.(67) is because $w_{i_s,s} \leq \frac{\alpha}{\|\boldsymbol{x}_{a_s}\|_{\boldsymbol{M}_{i_s,s}^{\prime-1}}} \leq \frac{\alpha}{\|\boldsymbol{x}_{a_s}\|_{\boldsymbol{M}_{i_s,t}^{\prime-1}}} \leq \frac{\alpha}{\|\boldsymbol{x}_{a_s}\|_{\boldsymbol{M}_{V_{i,t},t}^{-1}}}$ (since $\boldsymbol{M}_{V_{i,t},t} \succeq$

$\boldsymbol{M}'_{i_s,t} \succeq \boldsymbol{M}'_{i_s,s}$, $\boldsymbol{M}_{i_s,s}^{\prime-1} \succeq \boldsymbol{M}_{i_s,t}^{\prime-1} \succeq \boldsymbol{M}_{V_{i,t},t}^{-1}$, $\|\boldsymbol{x}_{a_s}\|_{\boldsymbol{M}_{i_s,s}^{\prime-1}} \geq \|\boldsymbol{x}_{a_s}\|_{\boldsymbol{M}_{i_s,t}^{\prime-1}} \geq \|\boldsymbol{x}_{a_s}\|_{\boldsymbol{M}_{V_{i,t},t}^{-1}}$), and
$\sum_{s\in[t]}|c_s| \leq C$.

Therefore, we have

$$\left\|\boldsymbol{\theta}_i - \hat{\boldsymbol{\theta}}_{V_{i,t},t}\right\|_2 \leq \frac{\sqrt{\lambda} + \sqrt{2\log(\frac{1}{\delta}) + d\log(1 + \frac{T_{V_{i,t},t}}{\lambda d})} + \alpha C}{\sqrt{\lambda_{\min}(\boldsymbol{M}_{V_{i,t},t})}}\,. \tag{68}$$

With Eq.(68), Eq.(61) and Eq.(56), together with Lemma 1, we have that for a normal user $i$, for any $t \geq T_0$, with probability at least $1 - 5\delta$ for some $\delta \in (0, \frac{1}{5})$

$$\left\|\tilde{\boldsymbol{\theta}}_{i,t} - \hat{\boldsymbol{\theta}}_{V_{i,t},t}\right\| \leq \left\|\tilde{\boldsymbol{\theta}}_{i,t} - \boldsymbol{\theta}_i\right\|_2 + \left\|\boldsymbol{\theta}_i - \hat{\boldsymbol{\theta}}_{V_{i,t},t}\right\|_2$$

$$\leq \frac{\sqrt{\lambda} + \sqrt{2\log(\frac{1}{\delta}) + d\log(1 + \frac{T_{i,t}}{\lambda d})}}{\sqrt{\lambda_{\min}(\tilde{\boldsymbol{M}}'_{i,t})}} + \frac{\sqrt{\lambda} + \sqrt{2\log(\frac{1}{\delta}) + d\log(1 + \frac{T_{V_{i,t},t}}{\lambda d})} + \alpha C}{\sqrt{\lambda_{\min}(\boldsymbol{M}_{V_{i,t},t})}}\,,$$

$$\tag{69}$$

which is exactly the detection condition in Line 7 of Algo.2.

Therefore, by the proof by contrapositive, we complete the proof of Theorem 5.

# F  Description of Baselines

We compare RCLUB-WCU to the following five baselines for recommendations.

- LinUCB[17]: A state-of-the-art bandit approach for a single user without corruption.
- LinUCB-Ind: Use a separate LinUCB for each user.
- CW-OFUL[12]: A state-of-the-art bandit approach for single user with corruption.
- CW-OFUL-Ind: Use a separate CW-OFUL for each user.
- CLUB[8]: A graph-based clustering of bandits approach for multiple users without corruption.
- SCLUB[21]: A set-based clustering of bandits approach for multiple users without corruption.

# G  More Experiments

## G.1  Different Corruption Levels

To see our algorithm's performance under different corruption levels, we conduct the experiments under different corruption levels for RCLUB-WCU, CLUB, and SCLUB on Amazon and Yelp datasets. Recall the corruption mechanism in Section 6.1, we set $k$ as 1,000; 10,000; 100,000. The results are shown in Fig.4. All the algorithms' performance becomes worse when the corruption level increases. But RCLUB-WCU is much robust than the baselines.

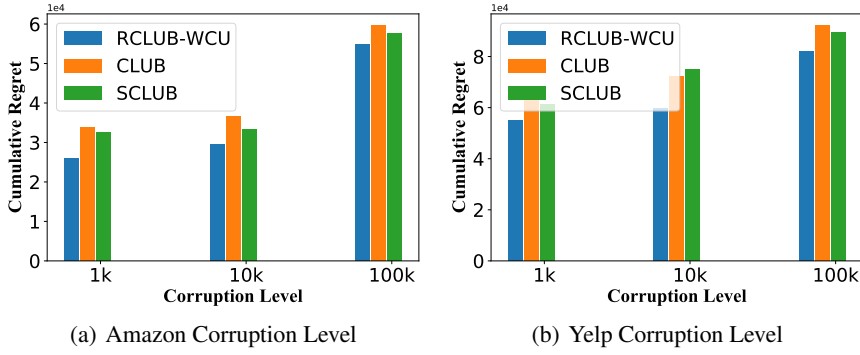

(a) Amazon Corruption Level          (b) Yelp Corruption Level

Figure 4: Cumulative regret in different corruption levels

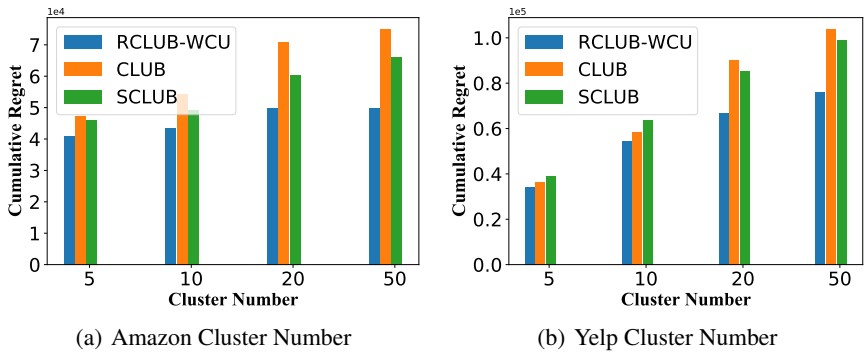

(a) Amazon Cluster Number          (b) Yelp Cluster Number

Figure 5: Cumulative regret with different cluster numbers

## G.2 Different Cluster numbers

Following [19], we test the performances of the cluster-based algorithms (RCLUB-WCU, CLUB, SCLUB) when the underlying cluster number changes. We set $m$ as 5, 10, 20, and 50. The results are shown in Fig.5. All these algorithms' performances decrease when the cluster numbers increase, matching our theoretical results. The performances of CLUB and SCLUB decrease much faster than RCLUB-WCU, indicating that RCLUB-WCU is more robust when the underlying user cluster number changes.

