# OpenReview forum: "Online Corrupted User Detection and Regret Minimization"
_NeurIPS.cc/2023/Conference — NeurIPS 2023 poster_

### Official Review · Reviewer_Lisv · 2023-07-07

**Soundness:** 2 fair
**Presentation:** 3 good
**Contribution:** 2 fair
**Rating:** 5
**Confidence:** 2

**Summary:**

This paper presents an important online learning problem named LOCUD to learn and utilize unknown user relations from disrupted behaviors to speed up learning and identify the corrupted users in an online setting. Also, the authors propose a novel bandit algorithm RCLUB-WCU, and devise a novel online detection algorithm OCCUD based on RCLUB-WCU’s inferred user relations. Extensive experiments demonstrate that the proposed methods can achieve superior performance over previous bandit algorithms and high corrupted user detection accuracy.

**Strengths:**

1. The paper is scientifically sound.
2. The clarity of the presentation is easy to follow.
3. Extensive experiments of the proposed methods have superior performance than other baselines.

**Weaknesses:**

1. The introduction section of the paper lacks sufficient emphasis on the motivation behind the proposed methods. The authors should provide a more comprehensive analysis of the current issues and challenges in the relevant fields, clearly indicating how their research work addresses and improves upon these challenges. This will help readers better understand the significance and contributions of the proposed methodology.
2. The abstract section should be more concise. It should effectively highlight the key innovations and improvements introduced by the proposed methodology.
3. The paper provides limited discussion and summary of the relevant literature. To strengthen the research methodology, the authors should include a more extensive review of existing research methods, along with an analysis of their strengths and weaknesses.
4. The experimental content is not enough to effectively prove the superiority of the method. It is suggested that the authors add more dimensional experiments and give a comprehensive analysis and explanation of the experimental results, thus making the conclusions in the paper more convincing.
5. The dataset used in the experiments is relatively small, which may limit the generalizability of the findings. It is recommended to supplement a large-scale real dataset for performance validation.


**Questions:**

Provided above.

**Limitations:**

Yes.

---

> ### Author Rebuttal · Authors · 2023-08-09
>
> # Responses to Reviewer Lisv
> Thanks for the positive comments and valuable suggestions for further improving our work. Our responses are listed below.
>
> ## 1. About the improving the writing and contents of the introduction, abstract, and related work sections:
>
> Thanks for giving these detailed and valuable suggestions on improving the writing and contents of our paper. For the introduction section, we will follow your advice to illustrate more on the motivations behind our proposed methods by providing more discussions on the current issues, challenges, and how we address these challenges. For the abstract section, we will highlight our contributions more concisely. For the related works, we will follow your suggestion to add more extensive discussions on existing relevant works with a more detailed analysis of their strengths and limitations in the Appendix.
>
> ## 2. About the experimental content:
> In our experiments, the number of datasets and the exploration of different settings are richer or comparable to related works [3, 7, 17, 18, 20, 24].
> In the previous works of clustering of bandits, [20] uses one synthetic dataset and two real-world datasets, and they do not conduct any studies in different settings. In [24], they employ one synthetic dataset and one real-world dataset with parameter study only on the synthetic dataset. In [18], they use one synthetic dataset and two real-world datasets and explore the influence of cluster numbers. In the previous works of bandits robust to corruption, [3] uses one synthetic dataset and one real-world dataset, and test both contextual and non-contextual settings only on the synthetic dataset; in [7], they use one synthetic dataset and two real-world datasets with study about the difference caused by attacking on a single context or more than one contexts. In the previous works of offline corrupted user detection, [17] uses two real-world datasets with one ablation study to explore how the components in their model influence the performance. Compared to the above previous works, our work includes the results on one synthetic dataset and three real-world datasets. And we add two additional experiments in two real-world datasets to observe the algorithm's performance with different corruption levels and cluster numbers. The number of datasets and the exploration of different settings are richer or comparable to related works.
>
> Apart from the experimental contents, we also give solid theoretical performance guarantees to prove the superiority of our proposed methods. We would add more experiments with more discussions and explanations. We would appreciate it if the reviewer could provide more specific suggestions on adding some experiments that could further improve our study.
>
>
> ## 3. About the size of the dataset for experiments:
>
> In the previous works of clustering of bandits and bandits with corruption, the sizes of datasets in most works are not very large and are close to ours [8, 18, 24, 32]. Therefore, following these works [8, 18, 24, 32], we extract a proportion of the large dataset to be the dataset used for experiments. We agree that performing experiments with a larger dataset would enhance the generalizability of our findings. Following your valuable advice, we have done some experiments on an enlarged dataset extracted from Yelp (the same rule of extraction, 20000 users and 20000 items, 10 times larger than the dataset used in our paper). The results are shown in Figure 4 and Table 3 in the global PDF. We can see our proposed algorithms also outperform baselines on this larger data set. We will conduct more experiments on larger datasets in a later version following your valuable advice.
>
>
> Finally, we thank the reviewer again for the positive feedback, the efforts in reviewing our paper, and giving valuable advice to further improve our work.

---

> > ### Comment · Reviewer_Lisv · 2023-08-18
> > **Read author's rebuttal**
> >
> > I appreciate the authors' detailed feedback on my comments. I would like to consider the rebuttal in my final comments.

---

> > > ### Author Response · Authors · 2023-08-19
> > > **Thanks for your review**
> > >
> > > Dear Reviewer Lisv,
> > >
> > > Thank you for reading our response and the positive review. We are grateful for your time and effort in the review and your valuable suggestions.
> > >
> > > Sincerely,
> > >
> > > Authors of Paper 5846

---

### Official Review · Reviewer_GoSQ · 2023-07-07

**Soundness:** 4 excellent
**Presentation:** 4 excellent
**Contribution:** 4 excellent
**Rating:** 7
**Confidence:** 2

**Summary:**

The paper considers the following bandit setup.

There are $u$ users organised into $m\ll u$ clusters. Each cluster has vector $\theta$ attached to it. On step $t$ the learner deals with a user uniformly selected from the pool and picks an arm $a$. If this is a bona fide user, the learner gets average reward $x'_a\theta$, where $x_a$ is the feature vector for the arm and $\theta$ is the vector for the cluster the user belongs to. There is a number of corrupted users though, who give reward $x'_a\theta + \eta + c$, where $\eta$ is noise and $c$ (a bounded quantity) is corruption.

We have a twofold problem of minimising the regret and identifying the corrupted users. The paper presents an algorithm based on a graph of connections between users built by observations of their behaviour. The regret upper bound is matched by a lower bound. It is shown that with high probability we identify the corrupted users correctly and after a while end up with correct clusters.

**Strengths:**

I think this is an interesting result and a strong guarantee. The explicit modelling of corrupted user behaviour may seem restrictive at first, but it covers many possibilities. The authors were very careful to relax all the requirements as much as possible (e.g., consider sub-Gaussian noise etc). The algorithm is intuitive.

**Weaknesses:**

No obvious weaknesses. The appeal of the result may be limited for some NeurIPS participants.

Some minor suggestions (not reflected in my evaluation of the paper):

1. I do not think quantifiers should be used as in

At $\forall t$, for any fixed unit vector $z$ ...

(Assumption 3.)

Using "every" here will not blow the volume of the paper out of proportions, but will improve readability.

2. Use $\verb!\left(...\right)!$ in formulas like (7) to get larger brackets.

3. I do not find the abbreviations used in the paper convenient and phrases such as

CW-OFUL-Ind outperforms LinUCB-Ind because it considers the corruption, but worse than RCLUB-WCU

easy to parse.

**Questions:**

None.

**Limitations:**

Yes.

---

> ### Author Rebuttal · Authors · 2023-08-09
>
> # Responses to Reviewer GoSQ
> We are very grateful for your strongly positive comments and appreciation. Below are our responses to your minor suggestions.
>
> ## 1. About the minor suggestions 1-2 on improving the format and readability:
> Thanks for giving these detailed suggestions. We will revise these two points following your valuable advice.
> ## 2. About the abbreviations:
> Thanks for your valuable suggestion. We will add a table to make it more convenient for the readers to find the abbreviations.
>
> Finally, thanks again for your positive feedback, time to review our paper, and detailed and valuable suggestions on improving our work.

---

### Official Review · Reviewer_HbaT · 2023-07-13

**Soundness:** 3 good
**Presentation:** 3 good
**Contribution:** 4 excellent
**Rating:** 6
**Confidence:** 3

**Summary:**

The authors introduce an online learning problem called LOCUD (Learning and Online Corrupted Users Detection from bandit feedback) in which the aim is to detect a small fraction of the overall users with corrupt behaviors; corrupt users occasionally perform undesirable actions, but otherwise mimic normal user behavior, making them challenging to detect.

The paper then proposes a framework that leverages the relations between users to form semantic clusters, and uses the clusters to identify corrupt users. Specifically, the authors propose RCLUB-WCU (Robust CLUstering of Bandits With Corrupted Users) to progressively prune a fully connected graph of users to clusters of connected components based on user interactions and preferences. Then, OCCUD (Online Cluster-based Corrupted User Detection) estimates a robust and non-robust estimation of each user's preferences, and identifies a user as corrupt if the gap between the two estimates exceeds a carefully designed threshold. OCCUD is repeatedly invoked within RCLUB-WCU to continually prune and refine the relational structure amongst the users.

Experiments one on synthetic dataset and three real-world datasets show the proposed approach is able to detect more corrupted users while achieving the least amount of regret over time than competing methods.

**Strengths:**

* The paper introduces a challenging but relevant problem of trying to identify corrupt users despite sporadic behavior in an online dynamic environment. This problem is especially relevant for sites like Amazon and Yelp which often contain users that exhibit corrupt behavior.

* The proposed approach of leveraging relational information between users to more effectively detect corrupt users is intuitive, and the experimental results suggest its effectiveness.

* Analyses and bounds related to regret are given for the RCLUB-WCU/OCCUD algorithm, with proofs provided in the Appendix.

* The paper is generally well-written and follows a logical progression.

**Weaknesses:**

* Experiments are performed on a small number of small datasets, potentially limiting the generalizability of the proposed approach. Performing experiments with a wider range of larger datasets would significantly benefit the claims made in the paper.

* No empirical runtime analysis is provided for the proposed approach or any of the competing methods. Runtime analysis can help practitioners decide what method is likely to work best for their particular problem.

* Only one baseline method is compared to the proposed approach for the AUC results (Table 1). Where are the results for the other methods?

* Additional experimental results including different corruption levels and number of clusters are provided in the Appendix, but those results (Figures 4 and 5) only shows regret, and only compares the proposed approach with two baseline methods. Can the authors provide results for the other baseline methods, as well as AUC results for these additional and potentially insightful experiments?

* Minor clarity improvements:
    * Use authors' names to cite previous work. "The work [5] proposes..." -> "Ding et al. (2022) propose...".
    * The legends in Figure 5 make the subplots hard to read. Figure 5 is also not color-blind friendly, consider adding markers or using different line styles for different methods.

**Questions:**

* Why are the dataset sizes so small (e.g., 1,400 users and 800 items for Amazon)? What are the original sizes of the datasets used in the experiments?

* Where are the error bars for Table 1, and Figures 4 and 5?

**Limitations:**

No, the authors have not addressed the limitations or the potential negative societal impact of their work.

---

> ### Author Rebuttal · Authors · 2023-08-09
>
> # Responses to Reviewer HbaT:
> Thanks for the positive comments and valuable suggestions for improving our work. We will revise the paper in the final version following your advice. Our responses are as follows.
> ## A. Responses to the Weakness:
> ### 1. About the small dataset:
> Please refer to the response to your first question later about the dataset.
>
> ### 2. About the runtime analysis
>
> Thanks for the valuable suggestion. We give the run time results with $T=1,000,000$ on our computer with CPU of Intel(R) Xeon(R) Gold 6240C CPU @ 2.60GHz, which has 36 cores and 2 threads in each core
>  in Table 1 in the global PDF. Overall there are no large differences since all the algorithms are computationally efficient.
>
> ### 3. About the baseline for online detection
> We compare OCCUD with one baseline when reporting the detection results in Table 1, because our paper is the first work to study the online corrupted user detection from bandit feedback, and no previous baselines exist. The baselines used to compare the online recommendations with RCLUB-WCU (in Figure 3) can not detect corrupted users, so they can not be compared with OCCUD. Also, the offline detection methods [34, 6, 17, 29] need to know all the user information in
> advance to derive the user embedding for classification, so they cannot be directly applied in online detection with bandit feedback.
> Therefore, we compare our OCCUD algorithm with a straightforward GCUD algorithm (which is proposed by us as a baseline to show that the design of OCCUD is non-trivial). Thanks for the valuable suggestion on adding more baselines. We add another straightforward baseline named GCUD2 which compares the non-robust estimators of clusters and users. We have added this GCUD2 baseline to Table 2 in the global PDF. The results also show the superior outperformance of our OCCUD algorithm. We will also consider adding more feasible baselines in a later version.
> ### 4. About experimental results with different corruption levels and number of clusters:
> Thanks very much for this valuable suggestion. In Figures 4 and 5 in the paper, we only compare with two baselines because these two baselines perform the best among all the baselines. The regret results of all the recommendation algorithms can be found in Figure 2 in the global PDF. The AUC results for online detection algorithms can be found in Figure 3 in the global PDF.
> These additional results also clearly show the good performance of our proposed algorithms. We will add these results in the final version following your valuable advice.
> ### 5. About the minor clarity improvements:
> Thanks for giving these detailed suggestions; we will do the revisions accordingly.
> ## B. Responses to the questions
> ### 1. About the size of the dataset:
> (1) In the previous works of clustering of bandits and bandits with corruption [8, 18, 24, 32], the sizes of datasets in most works are usually not very large and are close to ours. Therefore, following these works [8, 18, 24, 32], we extract a proportion of the large dataset to be the dataset used for experiments. We agree that performing experiments with a wider range of larger datasets would significantly benefit our claims. We also have some results on an enlarged dataset extracted from Yelp (the same rule of extraction, 20000 users and 20000 items, 10 times larger than the dataset used in our paper). The results are shown in Figure 4 and Table 3 in the global PDF. We can see our proposed algorithms also outperform baselines in this larger dataset.
> We will conduct more experiments on more and larger datasets in a later version following your valuable advice.
>
> (2) The original data sizes are: Movielens: 2,113 users and
> 10,197 items; Amazon: 1,429 users and 900 items; and Yelp: 1,987,929 users and 150,346 items.
>
> ### 2. About the error bars:
> Thanks for this valuable suggestion. Table 1, Figures 4 and
>  5 with error bars can be found in Table 2 and Figures 2 in the global PDF, respectively. We will add these error bars in the final version.
>
> Again, we thank the reviewer for the positive comments, the time spent on reviewing our paper, and the valuable advice on further improving our work.

---

> > ### Comment · Reviewer_HbaT · 2023-08-12
> > **Thank You**
> >
> > I thank the authors for addressing the majority of my concerns and have increased my score accordingly.

---

> > > ### Author Response · Authors · 2023-08-12
> > >
> > > Dear Reviewer HbaT,
> > >
> > > We are delighted to learn that we have successfully addressed your concerns. Your insightful feedback has played a crucial role in enhancing the quality of our work, and we sincerely appreciate your dedication in reviewing our paper thoroughly.
> > >
> > > Thank you once again for your time and constructive feedback.
> > >
> > > Sincerely,
> > >
> > > Authors of Paper 5846

---

### Official Review · Reviewer_wNgt · 2023-07-18

**Soundness:** 3 good
**Presentation:** 3 good
**Contribution:** 3 good
**Rating:** 7
**Confidence:** 2

**Summary:**

This research paper introduces an innovative method for learning and utilizing unknown user relations from disrupted behaviors to enhance the learning process and identify corrupted users in an online setting. To achieve this, a new bandit algorithm (RCLUB-WCU) is proposed, along with an online detection algorithm that leverages user relations inferred by RCLUB-WCU. The paper also presents a regret upper bound for RCLUB-WCU, which closely matches the lower bound with respect to T (the number of rounds) up to logarithmic factors and performs well even in degenerate cases. The experiments conducted on synthetic and real-world datasets demonstrate significant improvements in performance compared to previous bandit algorithms

**Strengths:**

1. The paper presents a novel application to learn unknown user relations in their preferences from potentially corrupted feedback. At the same time, the paper shows how to leverage the learned relations to speed up learning as well as adaptively detect the corrupted users online from bandit feedback. Overall I think this paper makes a significant contribution to the literature on online learning from corrupted feedback as well as detection of adversarial users in multi-user online learning setting.

2. Experiments on synthetic and real-world datasets clearly indicate  lower reward regret of proposed approach in comparison to five other baselines from the past literature on online clustering of bandits. At the same time, the algorithm is able to identify corrupted users with higher accuracy than a simple baseline that directly compares the robust-estimators of preference vectors of a user and its corresponding cluster.

3. The authors back up their results with theoretical analysis and guarantees on the performance of the proposed algorithm.

**Weaknesses:**

1. It seems to me that the performance of proposed approach could potentially be sensitive to the nature of underlying user relations in their preferences and the tightness of the detected clusters. For example, to my understanding, this algorithm might not work well if there are too many tight clusters. However, if one chooses too big clusters, then the solution might compromise on personalization as users in a loose cluster might not be represented accurately by a common preference vector defined for the cluster. The paper does not provide any discussion on the impact of cluster sizes on the algorithm.

2. The approach requires multiple parameters to be specified. (e.g. regularization parameter, confidence radius parameter, threshold parameter, edge detection parameter). There is no discussion provided on the sensitivity of the results to the choices of these parameters (in the main draft at least).

3. The results in Table 1 on detection of corrupted users are not very clear. To begin with, it's not clear what do numbers in the table indicate? I assume they indicate recall of true corrupted users. If so, where are the precision numbers? Where are F1 scores given that thre is class imbalance? Authors need to clearly indicate what they are measuring and also provide an explanation in case they aren't measuring both precision and recall. Also it would be nice to have comparison with multiple baselines. Further, authors could potentially enrich the results with other variants of the baseline, e.g. another baseline variant could simply compare the non-robust estimators of cluster and user. (I believe the current baseline compares the robust estimators of a user and its cluster).


**Questions:**

1. Page 9, lines 341-345: Authors claim that the performance of the proposed algorithm improves with proportion of corrupted users in the dataset. Seems like this argument is made based on the three datapoints (from three real-world datasets). This argument could be potentially strengthened by evaluating all algorithms while varying the number of corrupted users in one of the datasets.

2. Page 6: Line 220: It says that the threshold is carefully designed to handle the estimation uncertainty... How difficult/easy it is to tune this threshold? How critical this threshold is to the performance of the approach?

3. Pages 4-5, Lines 188-204: It seems like the subscript `t` is used to denote both timestamp (bandit round) as well as the cluster? Also I see subscript `s` to indicate the bandit iteration which should ideally be denoted by t ?

4. In Equation (2), are you considering data from all t -1 previous rounds? Or are you using iterative update and only include the data from previous round, i.e. (t -1)^{th} round?

**Limitations:**

Authors should cover any other limitations of this work in addition to potential limitations I highlighted above:

---

> ### Author Rebuttal · Authors · 2023-08-09
>
> # Responses to Reviewer wNgt:
> We appreciate the reviewer for the positive comments and valuable advice. We will incorporate the suggestions into the final version.
> ## A. Responses to the Weakness:
> ### 1. Impact of the cluster number and sizes:
> Thanks for your suggestion of adding discussions on the impact of the cluster number and sizes.
>
> As you noted, the performance of our algorithm is influenced by the cluster number $m$. As shown in Theorem 3, the regret upper bound of our algorithm increases with a larger $m$. The performance decreases with larger $m$ is inevitable for all clustering of bandits (CB) algorithms. The regret lower bound in Theorem 4 also supports this claim, theoretically indicating that no CB algorithms can avoid decreasing the performance with a larger $m$.
> Our empirical evaluations on different problem instances with different $m$ also validate this statement: (i) as detailed in Figure 5 and also mentioned in [18], the performance decreases with larger $m$ is observed for all CB algorithms. (ii) With different $m$, our algorithm outperforms the baselines (as detailed in Appendix G.2).
>
> For the second comment on the concern of ``big loose" clusters, our algorithm automatically learns the cluster sizes over time. Our algorithm clusters users adaptively during the learning process, and it will cluster all users correctly after some interactions (theoretically supported by Lemma 1), where users in each ground-truth tight cluster share the same preference vector. Therefore, the sizes of the clusters are not chosen by our algorithm but determined by the underlying problem instance. As shown in Theorem 3, the regret upper bound depends on the cluster number $m$ and does not depend on the sizes of the clusters. We will provide some discussions in the final version.
>
> ### 2. About the parameters:
> Following the publicly available code of the previous works on the clustering of bandits and bandits with corruption [12, 20], we set the regularization parameter $\lambda=1$, the confidence radius parameter $\beta=1.5$, the threshold parameter $\alpha=0.2$, and the edge deletion parameter $\alpha_1=1$. These are classic parameters in the previous clustering of bandits and linear bandits with corruption works [8, 12, 18, 20, 24]. To ensure the robustness, our algorithm does not introduce additional parameters compared to previous approaches [8,12,18,20,24]. And we use the same values for these parameters as in previous works to make fair comparisons with the baselines. We will specify these parameters in the experiment section and add more experiments to show the sensitivity of the parameters.
> ### 3. (a) About Table 1:
> Following the previous works on offline corrupted user detection [6, 17, 29, 34], we use AUC as the metric for online detection. We mentioned it in lines 349-351, and we will make it clearer by giving more descriptions and specifying AUC in the caption of Table 1.
> ### 3. (b) About more baselines:
> We compare OCCUD with one baseline when reporting the detection results in Table 1, because our paper is the first work to study the online corrupted user detection from bandit feedback, and no previous baselines exist. Thanks for the valuable suggestion on adding more baselines. We have done some experiments on the proposed baseline by simply comparing the non-robust estimators of cluster and user (GCUD2). The results (AUC) are shown in Table 2 in the global PDF. The results show the superior outperformance of our OCCUD algorithm. We shall consider adding more feasible baselines in a future version. Besides, please note that when we report the recommendation results (where there are more baselines applicable) in Figure 3, we compare our approach with several baselines.
>
> ## B. Answers to the questions:
> ### 1. About the relation between algorithm performance and proportion of corrupted users:
> Thanks for your valuable comment. We have done some experiments on the Movielens dataset with different proportions of corrupted users (10\%, 20\%, and 50\%). We compare our algorithm with CLUB and SCLUB (they perform the best among baselines on Movielens). The results are shown in Figure 1 in the global PDF. We can see that when there are more corrupted users, RCLUB-WCU's regret increases less than CLUB and SCLUB and gains a larger advantage. This is consistent with our argument in Line 341-345 of the paper. We will add these experimental results in the final version to strengthen the argument.
>
> ### 2. About the threshold:
> As mentioned in the above response, in this threshold, we do not introduce additional parameters to be tuned than previous works on the clustering of bandits and bandits with corruption [8, 12, 18, 20, 24], which ensures that our method is robust. And in empirical evaluations, we do not tune the threshold parameter $\alpha$ and the edge deletion parameter $\alpha_1$; we set them to be the same as the previous works to make fair comparisons [12, 20].
> ### 3. About the subscripts t and s:
> Yes, $t$ is used to denote both timestamps and clusters; we use $V_t$ to denote the cluster for user $i_t$ inferred by the algorithm at round $t$. We will change $s$ to $t$ and make the notations clearer following your comments.
> ### 4. About the iterative update:
> Eq.(2) shows that the estimated cluster vector is the solution of a weighted ridge regression considering data from all t -1 previous rounds. But in the algorithm, we can use the iterative update at each round that only includes the (t-1)-th data (Line 8 in Algo.1) thanks to the nice closed-form solution (Lines 194-195) of Eq.(2), which makes the algorithm computationally efficient.
>
> Thanks again for the reviewer's appreciation and the suggestions on improving our work.

---

> > ### Comment · Reviewer_wNgt · 2023-08-18
> > **Read author's rebuttal**
> >
> > I would like to thank authors for the detailed response to my original review. I have read author's rebuttal and will consider their arguments in my final rating.

---

> > > ### Author Response · Authors · 2023-08-18
> > > **Thanks for your review**
> > >
> > > Dear Reviewer wNgt,
> > >
> > > Thanks for reading our response and your positive review. We really appreciate your time and effort in the review and your valuable suggestions.
> > >
> > > Sincerely,
> > >
> > > Authors of Paper 5846

---

### Author Rebuttal · Authors · 2023-08-09

We sincerely appreciate all the reviewers for the positive comments, the time spent on reviewing our paper, and the valuable advice for improving our work. We have done some experiments for your reference. Please refer to the global PDF.

---

### Decision · Program_Chairs · 2023-09-21

**Decision:**

Accept (poster)

**Comment:**

The authors introduce a novel learning problem called for Online Learning and Online Corrupted Users Detection from bandit feedback. The paper proposes a new framework for leveraging the relations between users to form semantic clusters, and uses the clusters to identify corrupt users.

The problems as well as the a the approach are novel and interesting, and the paper is well written. I am happy to recommend for acceptance.